# Dynamics of CLIMP-63 S-acylation control ER morphology

Patrick A. Sandoz [1], Robin A. Denhardt-Eriksson[2], Laurence Abrami [1], Luciano A. Abriata [3,4], Gard Spreemann [5], Catherine Maclachlan[6], Sylvia Ho [1], Béatrice Kunz[1], Kathryn Hess [5], Graham Knott [6], Francisco S. Mesquita [1] ✉, Vassily Hatzimanikatis [2] ✉ & F. Gisou van der Goot [1] ✉

The complex architecture of the endoplasmic reticulum (ER) comprises distinct dynamic features, many at the nanoscale, that enable the coexistence of the nuclear envelope, regions of dense sheets and a branched tubular network that spans the cytoplasm. A key player in the formation of ER sheets is cytoskeleton-linking membrane protein 63 (CLIMP-63). The mechanisms by which CLIMP-63 coordinates ER structure remain elusive. Here, we address the impact of S-acylation, a reversible post-translational lipid modification, on CLIMP-63 cellular distribution and function. Combining native mass-spectrometry, with kinetic analysis of acylation and deacylation, and data-driven mathematical modelling, we obtain in-depth understanding of the CLIMP-63 life cycle. In the ER, it assembles into trimeric units. These occasionally exit the ER to reach the plasma membrane. However, the majority undergoes S-acylation by ZDHHC6 in the ER where they further assemble into highly stable super-complexes. Using super-resolution microscopy and focused ion beam electron microscopy, we show that CLIMP-63 acylation-deacylation controls the abundance and fenestration of ER sheets. Overall, this study uncovers a dynamic lipid post-translational regulation of ER architecture.

The endoplasmic reticulum (ER) is a complex multifunctional organelle that extends from the nuclear envelope to the cell periphery[1–3]. Based on morphological features, it is classically separated into three sub-compartments: the nuclear envelope, the rough ER, and the smooth ER. The rough ER consists of packed membrane sheets studded with ribosomes, concentrated in the perinuclear region. The smooth ER is formed by narrow tubular membranes arranged as a tentacular meshwork, of heterogenous density, that occupies the entire cytoplasm with a highly dynamic organization. Pioneering observations established that the relative abundance of ribosome-studded sheets and tubules varies between cell types and correlates with their function[4,5]. Sheets are the major site of synthesis of proteins destined for the secretory pathway and endomembrane system, and are very abundant in secretory cells[5,6], while tubules are thought to be involved in lipid biogenesis, calcium ion storage, and detoxification[7]. Over the past 25 years, the complex architecture of the ER has been shown to be orchestrated by specific membrane-shaping proteins[6,8–14], by proteins that coordinate contact with other cellular organelles[15–17], by proteins that control membrane fusion or fission[18,19] as well as by dynamic interactions with the cytoskeleton[20–24]. The local

[1]Global Health Institute, School of Life Sciences, EPFL, Lausanne, Switzerland. [2]Laboratory of Computational Systems Biotechnology, EPFL, Lausanne, Switzerland. [3]Laboratory for Biomolecular Modelling, Institute of Bioengineering, EPFL and Swiss Institute of Bioinformatics, Lausanne, Switzerland. [4]Protein Production and Structure Core Facility, School of Life Sciences, EPFL, Lausanne, Switzerland. [5]Brain Mind Institute, EPFL, Lausanne, Switzerland. [6]BioEM Facility, School of Life Sciences, EPFL, Lausanne, Switzerland. ✉e-mail: francisco.mesquita@epfl.ch; vassily.hatzimanikatis@epfl.ch; gisou.vandergoot@epfl.ch

concentration of different shaping proteins correlates with specific architectures and may theoretically explain the interconversion of the different ER morphologies, in a model that is reminiscent of phase diagrams[14]. A recent computational study suggested a primary role for the intrinsic curvature of membranes in controlling the formation of the tubular network as well as nanoholes within ER sheets[25]. A full mechanistic understanding of the formation and interconversion of sheets and tubules and the regulation thereof is however still lacking.

A key player in sheet formation is CLIMP-63 (cytoskeleton-linking membrane protein 63)[6]. CLIMP-63 is a type II membrane protein, with a short N-terminal cytosolic tail and a large C-terminal luminal domain[26]. The cytosolic tail has the ability to bind microtubules, thereby linking the ER to the cytoskeleton[27], and more specifically to centrosome microtubules[23]. The luminal domain has the capacity to multimerize through coiled-coil interactions[13,28]. It has been proposed that assembly occurs in *trans*, i.e., between CLIMP-63 molecules present in opposing membrane patches "across" the ER lumen, providing a mechanism to control the width of ER-sheets[6,29]. More recently, CLIMP-63 was found to coordinate the formation and dynamics of ER nanoholes by yet undetermined mechanisms[30,31]. A variety of studies have also reported that CLIMP-63 can act as a receptor for various ligands in a tissue-dependent manner, with significant clinical relevance[26,32–34], the most recent observation being a role for cell surface CLIMP-63 in inducing the secretion of von Willebrandt factor, after binding to the SARS-CoV-2 Spike protein[35]. Here we sought to better understand which mechanisms control the relative distribution of CLIMP-63 between the ER and the plasma membrane, and how, within the ER, CLIMP-63 is regulated to tune ER architecture.

We focused on the role of a specific post-translational lipid modification, S-acylation, which consists in the addition of a medium-length acyl chain to cytosolic cysteines, through the action of acyltransferases[36]. CLIMP-63 was found to be modified by the acyltransferases ZDHHC2[37] and ZDHHC5[33], which mostly localize to the plasma membrane and endosomal system[33,38]. Acylation was reported to control CLIMP-63 localization to specific plasma membrane domains and enhance its signalling capacity. Here, we investigated acylation of CLIMP-63 in the ER, where the bulk of the protein resides.

We combined various experimental methods (biochemistry, kinetic analysis, microscopy) with mathematical modelling of the enzymatic reactions, trafficking and degradation. We found that following synthesis in the ER, CLIMP-63 assembles into parallel homotrimeric units that can rapidly be S-acylated by the acyltransferase ZDHHC6, favouring their retention in the ER. De-acylation is mediated by the thioesterase APT2, and non-acylated trimers can exit the ER to reach the plasma membrane or instead be targeted for degradation. Therefore ZDHHC6/APT2-mediated cycles of S-(de)acylation coordinate the levels of CLIMP-63 between ER and plasma membrane. In parallel and within the ER, both acylated and non-acylated trimers form higher-order assemblies. This further stabilizes CLIMP-63 at the ER. Jointly, acylation and higher-order assembly render CLIMP-63 turnover in cells extremely slow, and thus control its abundance. Acylation of CLIMP-63 at the ER has therefore functional consequences: when amplified, causes loss of ER fenestration and a massive expansion of ER sheets, which can be counteracted by de-acylation. Our results reveal a dynamic ZDHHC6/APT2-mediated switch that directs ER morphology through the control of the ER-shaping protein CLIMP-63 cellular distribution.

## Results

### CLIMP-63 is present mainly in an acylated state in cells and in vivo

CLIMP-63 has been shown to undergo S-acylation on its sole cytosolic cysteine residue, Cys-100[33,37]. It has only one other cysteine, Cys-126, which is located on the luminal membrane boundary of the trans-membrane domain. To study CLIMP-63 S-acylation in-depth, we

generated a HeLa cell line stably depleted of the endogenous protein using shRNA (shCLIMP-63). We then optimised the expression of HA-tagged CLIMP-63, wild-type (WT) or mutant (C100A), in these cells by determining the amount of plasmid DNA required to reach near-endogenous protein expression levels and ensuring that the N-terminal tag did not affect WT CLIMP-63 cellular distribution (Supplementary Fig. 1a, b). Both WT and C100A distributed to the ER, as observed for endogenous CLIMP-63 (Supplementary Fig. 1b). Using this system, we confirmed that CLIMP-63 can undergo S-acylation by monitoring the incorporation of radioactive $^3$H-palmitate in WT CLIMP-63, but not in the C100A mutant (Fig. 1a). This is not specific to human CLIMP-63, since mouse CLIMP-63 is also S-acylated, on Cys-79, (Supplementary Fig. 1c, e).

In S-acylation, the lipid is linked to the protein via a thioester bond that can be broken in vitro using hydroxylamine. S-acylated proteins, such as CLIMP-63 (human or mouse) and calnexin, can be captured after hydroxylamine treatment using a method that has been termed Acyl-Rac (Supplementary Fig. 1d, e). Note that hydroxylamine treatment will break any thioester linkage, not only those involved in S-acylation. A variant of this method was used to estimate the proportion of S-acylated CLIMP-63. After cleavage with hydroxylamine the acyl chain is replaced with maleimide polyethylene glycol (mPEG - PEGylation) leading to a mass shift in SDS-PAGE gels. Following PEGylation, we found that the majority of WT CLIMP-63, but not the C100A mutant protein, migrated with a detectable mass change in a western blot analysis (Fig. 1b). Calnexin migrated as three bands, corresponding to S-acylation or not of its two cytoplasmic cysteines[39]. The mass of TRAPα was unaltered, as expected due to its lack of cytosolic cysteines (Fig. 1b).

For a more accurate quantification of CLIMP-63 S-acylation, we developed another variant of the Acyl-Rac assay, which involves an alkylation step with fluorescent iodoacetamide. This enables the detection of free, i.e., non-acylated, cysteines (Supplementary Fig. 1f). Cys-126 was mutated to Alanine to specifically quantify labelling of Cys-100. Only $12.7 \pm 0.05\%$ of CLIMP-63-C126A could be labelled without hydroxylamine treatment, (Fig. 1c), revealing that, in our system, more than 87% of CLIMP-63 is S-acylated at steady state. S-acylation was not restricted to cell lines (HeLa and retinal pigmented epithelial cells-Rpe1) as PEGylation performed on extracts of various mouse tissues indicated that CLIMP-63 is indeed mostly lipid-modified in vivo (Fig. 1d).

As its name indicates—cytoskeleton-linking membrane protein –, CLIMP-63 interacts with microtubules[20,23,27] via its N-terminal cytosolic tail. We investigated whether this interaction would influence S-acylation, which also occurs on the cytosolic domain. Incorporation of $^3$H-palmitate was not affected by microtubule-altering drugs, nor by mutations of the serine phosphorylation sites involved in microtubule binding (Fig. 1e, f). Consistently, the microtubule stabilizing drug paclitaxel/taxol had comparable effects on the distribution of CLIMP-63 WT and C100A mutant (Fig. 1g). Thus, S-acylation of CLIMP-63 occurs independently of interactions with microtubules.

Altogether these observations confirm that CLIMP-63 can be acylated on Cys-100 and show that in culture cells and in various mouse organs, the majority of CLIMP-63 molecules are lipid-modified, independently of their microtubule binding.

### ZDHHC6 S-acylates CLIMP-63 and controls its subcellular distribution

Two acyltransferases, ZDHHC2 and ZDHHC5, have been reported to modify CLIMP-63 and influence its cell surface distribution[33,37]. These enzymes localize primarily to the Golgi and plasma membrane[33,38], and possibly to endosomes or recycling endosomes. However, as CLIMP-63 localizes predominantly to the ER[6], additional, ER-localized ZDHHC enzymes must be involved to explain that more than 80% of the protein present in the cell is S-acylated. ZDHHC6 has been reported to modify various key ER proteins[39–41], prompting us to test its ability to

modify CLIMP-63. We generated a ZDHHC6 knockout (KO) cell line using the CRISPR-Cas9 system (Supplementary Fig. 2a, b). In these cells, $^3$H-palmitate incorporation into endogenous CLIMP-63, over a pulse of 2 h, was almost undetectable (Fig. 2a). While this observation indicates that ZDHHC6 is a major acyltransferase involved in CLIMP-63 S-acylation, we sought to evaluate whether in our hands we could also observe a contribution by ZDHHCs 2 and 5. We compared 2 h of $^3$H-palmitate incorporation into CLIMP-63, in control or cells silenced for ZDHHC 2, 5 or 6 (Fig. 2b, c). Silencing ZDHHC3, which localizes to the Golgi[42], was used as a negative control. Silencing of ZDHHC6 led to a decrease of $^3$H-palmitate incorporation of ~80%, silencing ZDHHC2 of ~30% and silencing of ZDHHC5 ~40% (Fig. 2b, c). These numbers do not add up to 100%. However, it is important to note that in $^3$H-palmitate incorporation experiments only non-acylated CLIMP-63 can acquire the label. Already acylated CLIMP-63, which, as we have shown above is the very vast majority, cannot. The amount of non-acylated CLIMP-63 available for acylation at steady state will vary when silencing one of the ZDHHC enzymes. This is likely the reason for the >20% changes observed when silencing ZDHHC2 and 5. In addition, the palmitoylation network is highly interconnected, both ZDHHCs and acyl protein thioesterases being themselves S-acylated, so silencing of one enzyme might somehow affect the activity of others.

Overexpression of individual ZDHHC enzymes had no significant increment on $^3$H-palmitate incorporation into CLIMP-63, consistent with the low proportion of non-acylated CLIMP-63 at steady state in control cells (Supplementary Fig. 2c, d).

Next, we monitored the interaction between the ZDHHC enzymes and CLIMP-63 using both co-immunoprecipitation experiments (Co-IP) and a proximity ligation assay, which allows the quantification of protein-protein interactions in the cellular environment[43]. CLIMP-63 co-precipitated with both ZDHHC2 and ZDHHC6, upon co-overexpression (Supplementary Fig. 2e). Proximity ligation, however, indicated a stronger association between CLIMP-63 and ZDHHC6, compared to ZDHHC2 (Fig. 2d, e), in line with the predominant ER-localization of CLIMP-63.

We investigated whether S-acylation of CLIMP-63 in the ER by ZDHHC6 could affect its abundance at the plasma membrane. Using a surface biotinylation assay, we confirmed that a proportion of CLIMP-63 is detected at the plasma membrane (Fig. 2f, g), as reported[33,37]. This population increased three-fold upon ZDHHC6 silencing (Fig. 2f, g), indicating that ZDHHC6 controls CLIMP-63 surface expression, presumably by trapping it in the ER. Consistent with an increased surface expression, the interaction between CLIMP-63 and ZDHHC2 was higher in ZDHHC6 KO than in control cells, as monitored by proximity ligation (Fig. 2d, e).

To rule out the general effects of ZDHHC6 silencing on biosynthetic trafficking, we monitored the distribution of CLIMP-63 C100A mutant, which also localizes mostly to the ER (Supplementary Fig. 1b). While the amount of C100A at the plasma membrane was far lower than that of WT CLIMP-63 in control cells (Fig. 2h), its surface abundance was insensitive to ZDHHC2 or 6 silencing (Fig. 2i and Supplementary Fig. 2f).

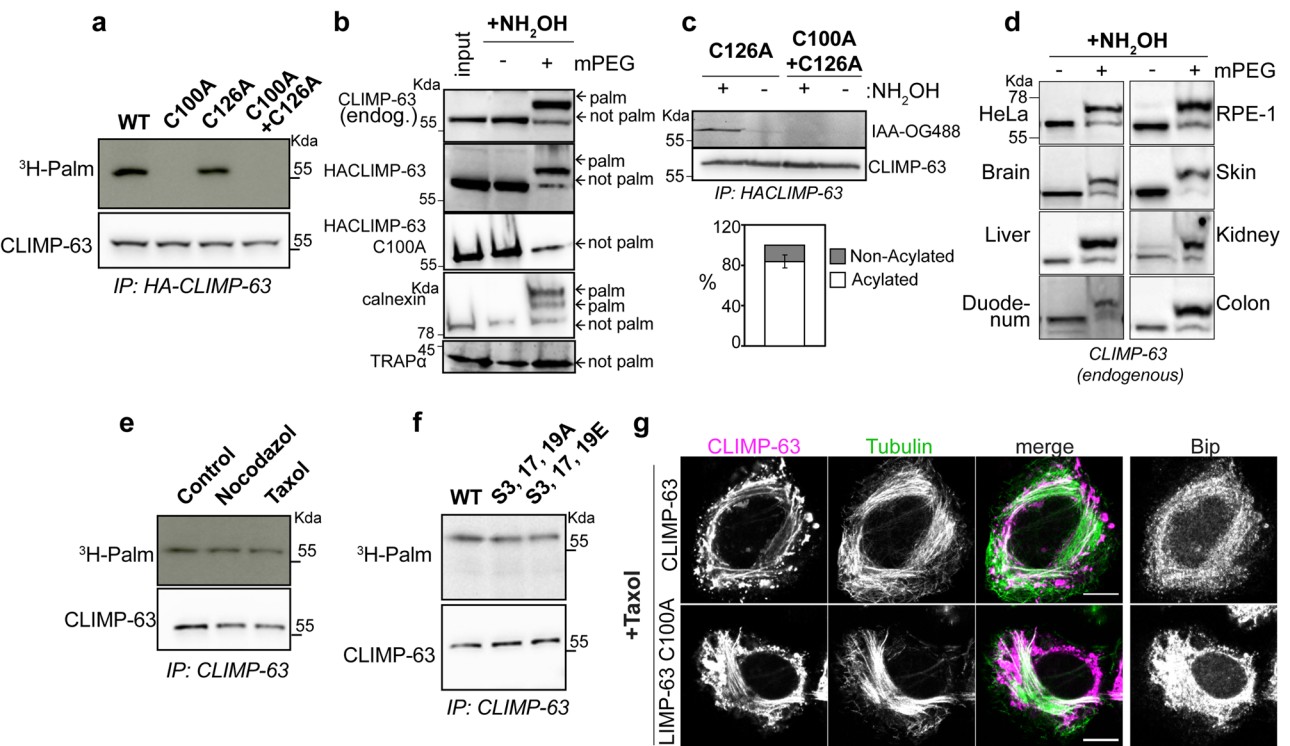

**Fig. 1 | The bulk of CLIMP-63 is S-palmitoylated in vitro and in vivo.**
**a** $^3$H-palmitate labelling of shCLIMP-63 HeLa cells expressing HA-CLIMP-63 WT, C100A, C126A or C100A-C126A mutants. Western blot and autoradiography show $^3$H-palmitate in HA-CLIMP-63 immunoprecipitation fractions (IP: HA-CLIMP-63). **b** PEG-labelling (+mPEG) was performed, or not (-mPEG) on endogenous CLIMP-63, transfected HA-CLIMP-63 WT or C100A mutant, endogenous calnexin and TRAP-alpha following treatment of HeLa lysates with hydroxylamine (NH2OH). The input corresponds to same amount of protein present in each condition. **c** Non-acylated fraction of CLIMP-63. Lysates from shCLIMP-63 HeLa cells expressing HA-CLIMP-63 C126A or C100A + C126A were treated or not with NH2OH and labelled with iodoacteamide-oregon-green-488 (IAA-OG488) as described in Supplementary

Fig. 1d. The amount of acylated CLIMP-63 was determined by comparing plus and minus NH2OH (Results are mean ± SD, $n = 4$ biologically independent experiments). **d** PEGylation of endogenous CLIMP-63, as in **b**, from lysates of different mouse tissues. **e, f** $^3$H-palmitate labelling of **e** HeLa cells mock-treated (Control) or pre-treated with nocodazole or Taxol or **f** shCLIMP-63 HeLa cells overexpressing CLIMP-63 WT or S3/17/19A or S3/17/19E serine mutants. Western blots show 3H-palmitate incorporation in IP fractions (IP: CLIMP-63). **g** Immunofluorescence of shCLIMP-63 HeLa cells expressing HA-CLIMP-63, treated with Taxol, and labelled for CLIMP-63 (Magenta), tubulin (Green) and ER marker Bip (Grey). Scale bar: 10 μm.

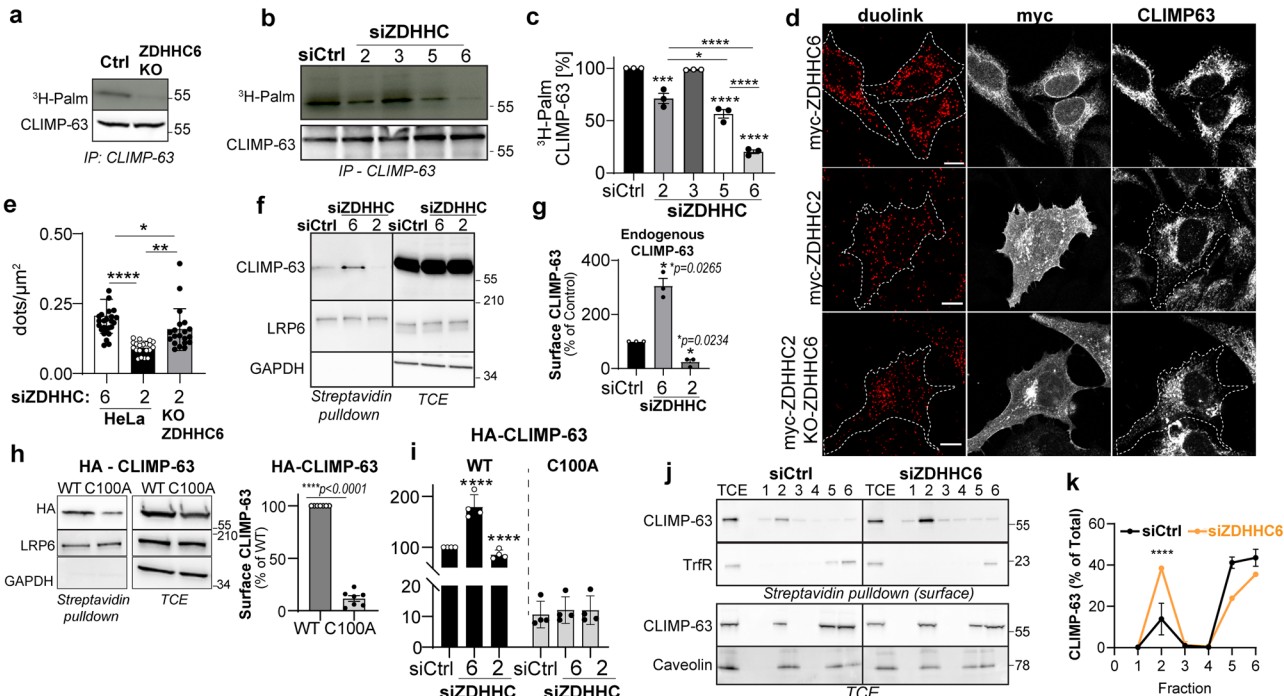

**Fig. 2 | ZDHHC6 palmitoylates CLIMP-63 and retains it at the ER. a** [3]H-palmitate labelling of CLIMP-63 immunoprecipitation fractions (IP) from control (Ctrl) or ZDHHC6 KO HeLa cells analysed by autoradiography and Western blot. **b** Same as in **a** but with HeLa cells transfected with control, siZDHHC2, siZDHHC3, siZDHHC5 or siZDHHC6. **c** Quantification of CLIMP-63 [3]H-palmitate in **b**. (n = 4). **d** Proximity ligation assay (duolink) probing endogenous CLIMP-63 in HeLa cells expressing myc-ZDHHC2 or myc-ZDHHC6, or in ZDHHC6 KO cells expressing myc-ZDHHC2. Scale bars: 10 μm. **e** Quantification of results in **d**, as duolink-dots per cell area (μm²) for 15 cells for each condition. **f** Western blot of surface biotinylated proteins and total cell extracts (TCE) from HeLa cells transfected with control, siZDHHC6 or siZDHHC2. LRP6 and GAPDH are positive and negative controls, respectively. **g** Quantification of surface CLIMP-63 in **f** (n = 3). **h** Analysis of surface proteins as in **f** from shCLIMP-63 cells transfected with HA-CLIMP-63 WT or C100A. (n = 7) **i** same

as in **h**, in shCLIMP-63 cells co-transfected with siCLIMP-63 plus control, siZDHHC6 or siZDHHC2 (n = 4). **j** Western blot of DRM fractionation of CLIMP-63 at the surface (top) or TCE (bottom) from HeLa cells transfected with control or siZDHHC6. Detergent-resistant membranes in fraction 2 are marked by caveolin; Transferrin Receptor (TrfR) is non-DRM surface control. **k** Quantification of CLIMP-63 in each fraction as a percentage of the sum of all fractions. p values compare CLIMP-63 in DRMs (fraction 2) (n = 3). For all graph data are mean ± SEM of the indicated biologically independent experiments). p values were obtained by **c, e, g** one-way ANOVA, **c, e** Tukey's (*p = 0.0336, ***p = 0.0003, *p = 0.0143, **p = 0.0021, ****p < 0.0001), **g** Dunnet's multiple comparison (*p = 0.0265, *p = 0.0234). **h** Paired two-tailed student's t test. and **i, k** two-way ANOVA Sydak's multiple comparison (****p < 0.0001).

At the cell surface, CLIMP-63 was shown to distribute to lipid raft-like domains in a S-acylation dependent-manner[33]. Association with detergent-resistant membranes—DRMs (Supplementary Fig. 2g) was used as a biochemical readout for raft association[44]. Membrane nanodomains are indeed resistant to solubilization with cold detergent, and therefore float in Optiprep™-density gradients, along with established markers of such domains (e.g. caveolin-1) (Supplementary Fig. 2g). We could confirm that a small population of endogenous CLIMP-63 (~14%) associated with DRMs (Fig. 2j, k), which was not observed for the C100A mutant (Supplementary Fig. 2h, i), consistent with previous observations[33,37]. In combination with surface biotinylation, we demonstrated that cell surface CLIMP-63 exclusively distributed to DRMs (Fig. 2j). Silencing ZDHHC6 increased CLIMP-63 plasma membrane localization (Fig. 2g, i), and presence in DRMs (Fig. 2j, k) further supporting a role of ZDHHC6 in controlling plasma membrane levels of CLIMP-63.

Altogether, these observations show that S-acylation by multiple ZDHHC enzymes controls the subcellular distribution of CLIMP-63: the majority of CLIMP-63 undergoes S-acylation in the ER by ZDHHC6 leading to ER retention, a proportion of non-acylated CLIMP-63 exits the ER and undergoes acylation by ZDHHC2/5 later in the secretory pathway or in the plasma membrane - endosomal system. This acylation is important for its sustained presence at the cell surface, within lipid nanodomains[33].

**CLIMP-63 S-acylation can be reversed by acyl protein thioesterase 2**
We next examined the kinetics of CLIMP-63 S-palmitoylation and depalmitoylation. [3]H-palmitate incorporation increased gradually over 6 h (Fig. 3a, Supplementary Fig. 3a). [3]H-Palmitate turnover was monitored by a pulse-chase approach, where a 2 h pulse was followed by different periods of chase in label free medium. Approximately 50% of CLIMP-63-bound [3]H-palmitate was released within 30 min (Fig. 3b, Supplementary Fig. 3b), indicative of rapid depalmitoylation. However, ~20% of CLIMP-63 remained radioactively labelled even after a 5 h chase (Fig. 3b), indicating the presence of longer-lived palmitoylated-CLIMP-63 species. Silencing ZDHHC2 had no significant effect on palmitate turnover, whereas silencing ZDHHC6, despite drastically reducing CLIMP-63 palmitoylation (Fig. 2b, c), allowed the detection of a minor population of palmitoylated-CLIMP-63 with a slower depalmitoylation rate (Fig. 3c).

Deacylation is mediated by Protein Acyl Thioesterases (APTs)[36]. We tested the potential involvement of APT1 and APT2. The [3]H-palmitate turnover was insensitive to APT1 silencing, but significantly delayed upon APT2 siRNA (Fig. 3d, Supplementary Fig. 3c). The same observations were made when using ML348 and ML349, specific inhibitors of APT1 and APT2 respectively (Fig. 3d, Supplementary Fig. 3d). Consistent with these results, ectopically expressed APT2 and the catalytic inactive mutant S122A could be co-immunoprecipitated with endogenous CLIMP-63 (Fig. 3e). We next investigated the effect of ML349 on the amount of CLIMP-63 at the cell surface. ML349 led to an almost fourfold increase of endogenous CLIMP63 at the plasma membrane (Fig. 3f),

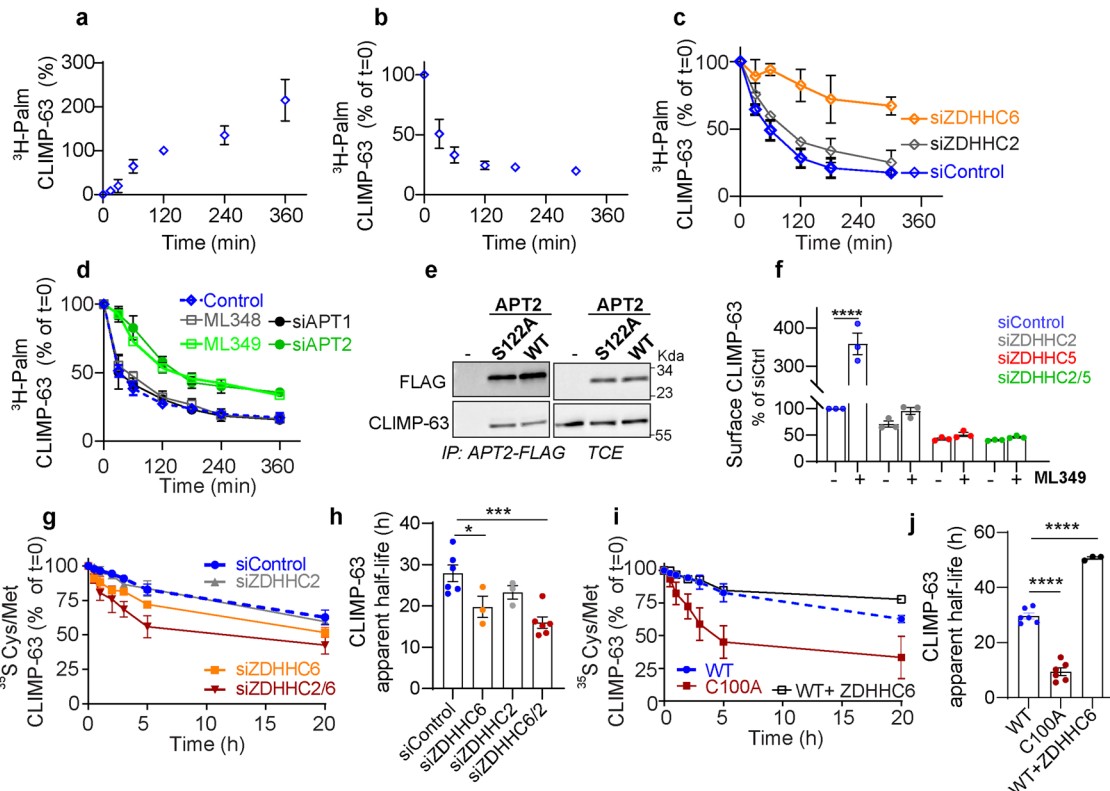

**Fig. 3 | Cycles of De/Palmitoylation of CLIMP-63 by APT2 and ZDHHC6 control its turnover and localisation. a** Quantification of [3]H-palmitate incorporation into CLIMP-63. Values were normalized to the population at 2 h as 100%. Results are mean ± SD, n = 3. **b** Quantification of [3]H-palmitate decay from CLIMP-63 after a 2 h pulse of [3]H-palmitate labelling followed by the indicated periods of chase time. Values were normalized to the initial population (t0) as 100% (results are mean ± SD, n = 5. **c, d** Quantification of [3]H-palmitate turnover from CLIMP-63 as in **b**: **c** HeLa transfected with control (Ctrl), siZDHHC2 or siZDHHC6 or **d** HeLa transfected with siAPT1 or siAPT2, or treated with specific APT inhibitors ML348 or ML349. Results are mean ± SD, n = 3. **e** Co-immunoprecipitation of endogenous CLIMP-63 with overexpressed APT2-FLAG WT or S122A in Rpe-1 cells. **f** Quantification of CLIMP-63 population at the cell surface in HeLa cells transfected with control, siZDHHC2, siZDHHC5 or siZDHHC2/5 mock-treated (-) or treated with

ML349 for 4 h before surface biotinylation (results are mean ± SEM, n = 3, p values compare surface CLIMP-63 ± ML349. **g–j** CLIMP-63 apparent decay after 20 min pulse of [35]S metabolic labelling, followed by the indicated periods of chase time. Each sample value was normalized to the initial population (t0) as 100%. Results are mean ± SD. **h, j** Apparent half-lives were extracted from the individual experiments using non-linear regression with one-phase decay. Results are mean ± SEM; *p = 0.0372, ***p = 0.0005, ****p < 0.0001 obtained by one-way ANOVA, Dunnet's multiple comparison. **g, h** Cells were transfected with siZDHHC2, siZDHHC6 (n = 3) and siZDHHC2/6 or control siRNA (n = 6), and **i, j** shCLIMP-63 cells were transfected with either HA-CLIMP-63 WT or C100A (n = 6), or HA-CLIMP-63 + ZDHHC6-myc together (n = 3). Unless indicated otherwise, all data is represented as mean ± SEM of independent biological experiments.

while as expected ML349 did not affect the amount of surface C100A mutant (Supplementary Fig. 3e). We have previously shown that inhibiting APT2 leads to rapid degradation of ZDHHC6[40]. The observed ML349-induced increase of CLIMP-63 at the cell surface could thus be partly due to an indirect effect on ZDHHC6. We therefore repeated the experiment in ZDHHC6 KO cells. ML349 still had an effect on endogenous surface CLIMP63 (Supplementary Fig. 3f), although lower, arguing for an effect of ML349 on ZDHHC6. The effect of ML349 on the surface expression of CLIMP-63 was essentially lost when ZDHHC2, 5 or both were silenced in Ctrl cells (Fig. 3f) and in ZDHHC6 KO cells (Supplementary Fig. 3f). These observations indicate that surface CLIMP-63, once S-acylated by ZDHHC 2 or 5, can undergo de-acylation by APT2, and that this de-acylation at the plasma membrane leads to a decrease of surface CLIMP-63, presumably due to retrieval of non-acylated CLIMP-63 by endocytosis[33].

S-acylation has been reported to impact the turnover rate of various proteins[36,39,40,45,46]. We therefore studied the effect of S-acylation on CLIMP-63 stability using [35]S Cys/Met metabolic pulse-chase experiments (Fig. 3g–i). After a 20 min labelling pulse, endogenous CLIMP-63 was slowly degraded, following a somewhat biphasic kinetic (see below, mathematical modelling section), which displayed an apparent half-life ($t_{1/2}$) of ≈30 h (Fig. 3g, h). Silencing ZDHHC6 accelerated the decay ($t_{1/2}$ = 22 h), whereas ZDHHC2 depletion had

little effect ($t_{1/2}$ ≈ 27 h) (Fig. 3g, h). Silencing both enzymes, however, had a pronounced effect ($t_{1/2}$ ≈ 14 h) (Fig. 3g, h), confirming that ZDHHC6 acts upstream from ZDHHC2. We also monitored the turnover of the S-acylation deficient C100A mutant and compared it to ectopically expressed WT CLIMP-63 (Fig. 3i). C100A was dramatically less stable ($t_{1/2}$ ≈ 7 h) than WT (Fig. 3i, j) and the half-life was insensitive to ZDHHC6 knock out (Supplementary Fig 3g). The mutation of Cys-100 had a stronger effect than silencing both ZDHHC6 and ZDHHC2, indicating that either ZDHHC5 (reported during the course of this study to modify CLIMP-63 at the plasma membrane[33]) or/and residual ZDHHC2/6 palmitoylating activity after silencing, still stabilised CLIMP-63 in our setting. Finally, ZDHHC6 overexpression resulted in a strong stabilization of CLIMP-63 (apparent $t_{1/2}$ ≈ 56 h) (Fig. 3i, j). Thus S-acylation, mediated by ZDHHC6, 2 and presumably 5, strongly influence the turnover rate of CLIMP-63.

**Trimerization and higher-order assembly of CLIMP-63**
To better understand how the subcellular distribution and turnover rate of CLIMP-63 is controlled by cycles of acylation and deacylation, we generated a conceptual computational representation of the system using mathematical modelling. Our initial model was composed of five CLIMP-63 species: acylated or non-acylated monomers (or Elementary units-E) in the ER ($E^0_{ER}$, $E^1_{ER}$, where 0 and 1 superscripts

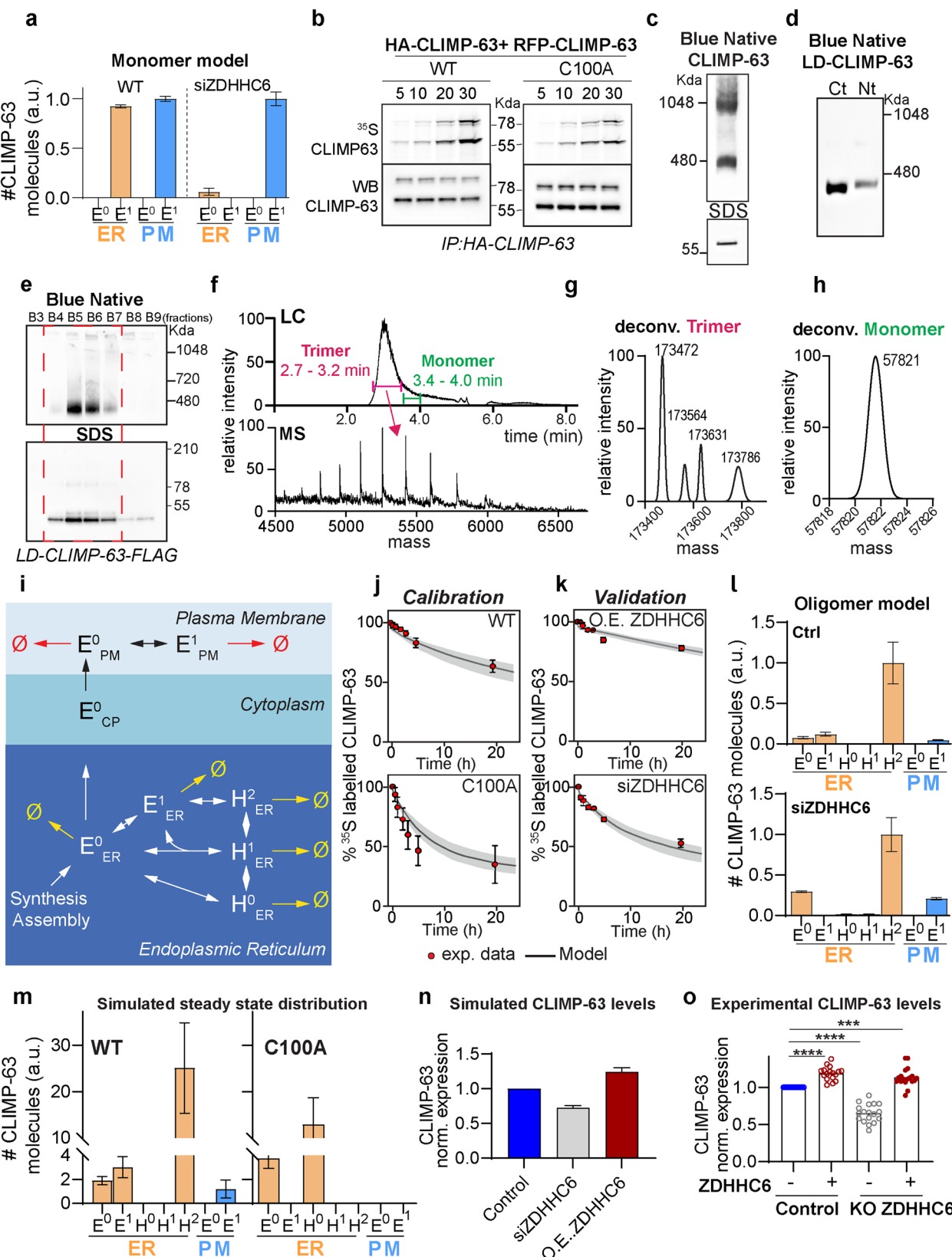

indicate whether the S-acylation site is free or modified), and at the plasma membrane (PM) ($E^0_{PM}$, $E^1_{PM}$) and a non-acylated transport intermediate ($E^0_{CP}$). This model properly captured our pulse-chase experiments (Supplementary Fig. 4a), but predicted equal distribution of CLIMP-63 between the ER and the plasma membrane, with a complete relocation to the plasma membrane upon ZDHHC6 depletion

(Fig. 4a). This was inconsistent with the experimental observations, where the bulk of CLIMP-63 resides in the ER, even in the absence of ZDHHC6 or upon mutation of Cys-100 (Supplementary Fig. 1b). The inability of the model to adequately capture the system highlighted the absence of a key mechanistic element to understand the subcellular distribution of CLIMP-63.

**Fig. 4 | Modelling of CLIMP-63 palmitoylation dynamics. a** Simulation of the steady-state distribution of CLIMP-63 as predicted by the monomeric (M) model, with or without ZDHHC6 silencing. 0 and 1 superscripts indicate free or S-acylated CLIMP-63. **b** HA-CLIMP-63 and RFP-CLIMP-63 co-immunoprecipitation and 35S Cys/Met metabolic labelling of shCLIMP-63 HeLa cells expressing (WT or C100A mutant). Western blot analysis shows equivalent Co-IP of newly synthesised HA- and RFP-tagged CLIMP-63. **c** Western blot analysis of CLIMP-63 migrated on Blue Native or SDS-PAGE gels. **d** Blue Native gel analysis of shCLIMP-63 HeLa cells expressing CLIMP-63 luminal domain (LD) mutant with C- or N-terminal FLAG tag. **e** Western blot analysis of chromatography fractions of LD-CLIMP-63-FLAG on Blue Native or SDS-PAGE gels. Red square indicates fractions used for subsequent mass spectrometry analysis. **f** Intact mass LC-MS analysis under native-like conditions (Raw mass spectrum corresponding to LC peak between 2.7 and 3.2 min, magenta) of purified LD-CLIMP-63-FLAG. **g, h** Deconvolved mass spectra of the indicated LC peaks revealed: **g** a molecular mass between 173472 & 173786 Da, CLIMP-63 luminal trimers and **h** a mass of 57821 Da, monomers (green peak). **i** Schematic description of CLIMP-63 species and localization. E-Elementary trimer units, H-higher-order assemblies, and 0, 1 and 2 superscripts indicate zero, single, and double S-acylation

of E within H. Acylation is catalysed by ZDHHC6 in the ER, and by ZDHHC2/5 at the plasma membrane. De-acylation is catalysed by APT2, both at the ER and at the plasma membrane. **j** Calibration and **k** validation of the model. The solid line represents the median of 100 simulated parameter sets; the shaded grey interval is defined by the 1st and the 3rd quartile; red points correspond to experimentally retrieved data points depicted in Fig. 3. **l** Simulation of the steady-state distribution of CLIMP-63 predicted by the oligomeric model (E + H), with and without ZDHHC6 silencing. **m.** Steady-state distribution of the different CLIMP-63 species as predicted for CLIMP-63 WT and C100A. **n** In silico (model) prediction of the total level of CLIMP-63 under control conditions (blue), ZDHHC6 silencing (grey), or ZDHHC6 overexpression (red). **o** Quantification of Western blot analysis of endogenous CLIMP-63 levels in control HeLa or ZDHHC6 KO cells overexpressing or not ZDHHC6. Values were normalised to the CLIMP-63 levels in control condition as 1. Results are mean ± SD and each data point corresponds to one biologically independent experiment−$n > 17$; $p$ values were obtained by one-way ANOVA, Dunnet's multiple comparison (***$p = 0.0004$, ****$p < 0.0001$). All simulation data sets represent the median, and error bars the first and third quartile or SD (bar charts) through the simulation of $n = 100$ models.

---

We hypothesized that the missing element could be multimerization of CLIMP-63[13,28,29]. Information on CLIMP-63 oligomerization is limited, prompting us to further analyse it. First, we verified that CLIMP-63 can self-assemble by performing Co-IP experiments using shCLIMP-63 cells co-expressing HA-CLIMP-63 and RFP-CLIMP-63 (Fig. 4b and Supplementary Fig. 4b). Co-IP in combination with 35S Cys/Met metabolic labelling showed that CLIMP-63 monomers interact and assemble rapidly following synthesis (Fig. 4b), irrespective of S-acylation. Blue-NATIVE PAGE revealed 2 prominent CLIMP-63 bands, with apparent molecular weights of ~480 and 1048 kDa (Fig. 4c), and no band corresponding to the monomer size.

To precisely study the stoichiometry of CLIMP-63 complexes, we generated a construct to express a soluble ER luminal domain (with a predicted mass of ~58 kDa) with a N-terminal signal sequence for targeting to the ER lumen and a His-FLAG tag, either at the C-terminus or at the N-terminus, for purification. The protein was secreted by the cells and could be purified from the culture medium. Blue-Native PAGE showed that this CLIMP-63 luminal domain migrates predominantly as a single species, just below the 480 kDa marker (Fig. 4d, e). The C-terminal-tagged luminal CLIMP-63 domain was further analysed by Intact Protein Liquid Chromatography Mass Spectrometry (LC-MS). We almost exclusively detected a complex of approximate 173.4–173.8 kDa (Fig. 4f, g), which would correspond to trimers of the luminal domain, and very small amounts of a ~57.8 kDa protein (Fig. 4h), likely corresponding to monomers. Exact molecular mass determination under denaturing conditions and shotgun proteomics (Supplementary Fig. 4c, d) confirmed that our samples contained solely the luminal domain of CLIMP-63.

Altogether these observations indicate that full length CLIMP-63 assembles into elementary trimeric units, which can further assemble into higher ordered assemblies, based on the migration in Blue Native PAGE, possibly dimers of trimers or trimers with other proteins. Since the vast majority of CLIMP-63 is in the ER, the migration pattern of CLIMP-63 in Blue Native gels, with bands at 480 and 1048 kDa, reports on the ER population and thus indicates that both trimers and higher order assemblies are present in the ER. The fact that CLIMP-63 forms trimers, rather than dimers, argues for a parallel assembly of monomers, through the proposed coiled-coil interactions.

## Mathematical model of CLIMP-63 assembly, trafficking and turnover

With this additional information on the quaternary assembly of CLIMP-63, we could generate a more complex model. A diagram of the model is shown in Fig. 4i, with a full description in Supplementary Information.

In the ER, CLIMP-63 is present either as elementary (E) units, the trimer, or a higher-order (H) CLIMP-63 assemblies (Fig. 4i). We tested different sizes of assemblies, but this did not change the behaviour of the model. Therefore, H was modelled as a dimer of elementary units, consistent with the Blue Native analysis. For simplicity, all the S-acylation reactions of E were grouped into one, leading to five possible species in the ER: $E^0$, $E^1$, $H^0$, $H^1$ (in which only one E is acylated) and $H^2$ (both Es are acylated). Only $E^0$ was given the ability to be transported to the plasma membrane, based on our observation that only non-acylated CLIMP-63 exits the ER. To incorporate the time delay corresponding to the transport of CLIMP-63 from the ER to the plasma membrane, we included a "cytoplasmic" $E^0$ ($E^0_{CP}$) species. At the cell surface, we include two species $E^0$ and $E^1$ (Fig. 4i).

Acylation was modelled as an enzymatic reaction catalysed by ZDHHC6 in the ER, and by ZDHHC2/5 at the plasma membrane. De-acylation was modelled as an enzymatic reaction catalysed by APT2, and was possible both at the ER and at the plasma membrane. No reactions other than transport were implemented in the "cytoplasmic" compartment. Acylation and deacylation were modelled with total quasi-steady-state assumption (tQSSA) enzyme kinetics. All other reactions were modelled with mass-action rate laws. Each species was given its own first order degradation rate constant.

A subset of the data from our pulse-chase experiments was used to calibrate the model (Fig. 4j and Supplementary Fig. 4e). A heuristic optimization method generated 100 parameter sets that satisfactorily fitted all the calibration experiments. The 100 parameter sets with the best fits were subsequently used to predict the results of a second, independent, set of experiments, i.e., validation experiments. All the predictions fitted the experimental data (Fig. 4a, k and Supplementary Fig. 4f). The introduction of higher-order complexes, H, in the ER allowed the correct prediction of the subcellular distribution, with the vast majority of CLIMP-63 residing in the ER, both in control and ZDHHC6 siRNA conditions (Fig. 4l).

The model was next used to predict the distribution of the different CLIMP-63 species. $H^2_{ER}$ was predicted to be by far the most abundant form (Fig. 4l), even upon silencing of ZDHHC6. This is not surprising because acylation is not required for higher-order assembly (Fig. 4b) and secondly, since silencing is not a knock out, a 10% residual ZDHHC6 activity was assigned in the model to the siZDHHC6 condition. This was sufficient to produce $H^2_{ER}$ over time (Supplementary Fig. 5a). Consistent with our experimental observations (Fig. 2f, g), the model predicted an increase of CLIMP-63 at the cell surface in ZDHHC6 silenced cells (Fig. 4l Supplementary Fig. 5a). There, it was predicted to be in the $E^1_{PM}$ form, consistent with the experimental observation that ZDHHC2/5-mediated acylation of CLIMP-63 increases

its abundance at the cell surface (Figs. 2f, g, 3f and Supplementary Fig. 3f).

We also modelled the steady state distribution of the C100A mutant as compared to WT. Again, consistent with the experimental data, C100A was still mostly in the H-form at the ER (Fig. 4m) and undetectable at the cell surface, as observed by surface biotinylation (Fig. 2h).

We next calculated the impact of ZDHHC6 activity on overall CLIMP-63 cellular abundance. Overexpression of ZDHHC6 was predicted to increase total CLIMP-63 levels by 30% (Fig. 4n), whereas silencing ZDHHC6 decreased CLIMP-63 levels by 32% (Fig. 4n). Again, these predictions could be confirmed experimentally. CLIMP-63 levels were 30% lower in ZDHHC6 KO cells and 20% higher in ZDHHC6 overexpressing cells (Fig. 4o).

Finally, we performed a global sensitivity analysis to determine the parameters that contribute the most to the accurate calibration of the model. These, in turn, reflect the biological constraints that govern CLIMP-63 levels and cellular distribution (Supplementary Fig. 5b, c). Three top parameters that emerged are: the catalytic rate of ZDHHC6: (kcat6); the Michaelis–Menten constant (KM) of ZDHHC6-mediated acylation (KM6) and the rate at which CLIMP-63 exits the ER (knpER_CP) (Supplementary Fig. 5b, c). The two next parameters were: the kinetics of the formation of H (kdim) and the degradation of non-acylated trimers in the ER $E^0_{ER}$ (kdC0ER). This sensitivity analysis indicates that the life-cycle of CLIMP-63 is most significantly controlled by two processes: its acylation by ZDHHC6 and its assembly into higher-order structures. This in turn controls both its exit from the ER and turnover rate in the ER. Thus, two mechanisms mediate ER retention of CLIMP-63: acylation of trimers and their higher-order assembly.

## Higher-order assembly of CLIMP-63 protects the protein from depalmitoylation

A powerful aspect of mathematical modelling is the possibility of interrogating it to obtain information that may not be readily accessible experimentally. For instance, $^{35}$S Cyst/Met metabolic pulse-chase kinetics can be deconvoluted to determine the evolution of the individual CLIMP-63 species over time (Fig. 5a). The model predicts that following synthesis, elementary CLIMP-63 units ($E^0_{ER}$) are formed. These rapidly undergo S-acylation ($E^1_{ER}$) and only then assemble into higher-order complexes ($H^2_{ER}$), as suggested by the predicted absence of $H^0_{ER}$. S-acylation is not required for higher-order assembly, but since $E^1_{ER}$ is predicted to be 1.6 times more abundant than $E^0_{ER}$, formation of $H^2_{ER}$ is more likely to occur than that of $H^0_{ER}$. Thus, 20 h after CLIMP-63 synthesis, $H^2_{ER}$ is the major species (Fig. 5a, WT). A minor population of $E^0_{ER}$ exits the ER to reach the PM, where it exclusively accumulates in the acylated form $E^1_{PM}$. The deconvolution of $^{35}$S Cyst/Met decay curves is much simpler for the S-acylation deficient C100A mutant: $E^0_{ER}$ forms and converts into higher-order $H^0$ complexes (Fig. 5a, C100A).

The model also allowed estimating palmitoylation and depalmitoylation rates of the various CLIMP-63 species (Fig. 5b). $E_{ER}$ was predicted to undergo rapid palmitoylation as well as depalmitoylation (Fig. 5b). In contrast, $H_{ER}$ displayed minimal acylation and deacylation (Fig. 5b). These predictions suggest that the $^3$H-palmitate pulse chase experiments (Fig. 3b) were capturing the depalmitoylation of elementary units, and thus, that only $E_{ER}$ were undergoing significant palmitoylation during the 2 h pulse. The model indeed predicts that after 2 h labelling, the $^3$H-palmitate-labelled population is 78% $E^1_{ER}$ and only 15% $H^2_{ER}$ (Fig. 5c). These proportions could be shifted by increasing the pulse period. After a 20 h pulse, 65% of the labelled population was predicted to be $H^2_{ER}$ (Fig. 5c). As the percentage of $H^2_{ER}$ at the end of the pulse period increased, $^3$H-palmitate decays were predicted to slow down (Fig. 5d), as we could validate experimentally (Fig. 5e). Thus, our mathematical model, supported by the experimental data, showed that CLIMP-63 trimers rapidly undergo acylation

in the ER, but are vulnerable to de-acylation. Higher-order assembly of CLIMP-63 however protects it from de-acylation.

## Jointly S-acylation and higher-order assembly control CLIMP-63 stability

We next used the model to infer the half-lives of the different CLIMP-63 species, parameters that are not easily ascertained experimentally. Most species were predicted to have very similar half-lives of ~5 h. One notable exception was $H^2_{ER}$, at above 80 h (Fig. 5f), consistent with it being $H^2_{ER}$ is the most abundant CLIMP-63 species in the cell (Fig. 4l). We however sought to confirm this prediction experimentally. We generated a fusion protein of CLIMP-63 with an N-terminal SNAP tag to fluorescently label fully folded proteins and monitor their decay with time[45]. Consistent with the prediction, SNAP-CLIMP-63 did not undergo significant degradation over 24 h (Fig. 5g).

Analysis of the half-lives of CLIMP-63 species indicates that individually, S-acylation or higher-order assembly do not stabilize CLIMP-63 in the ER ($E^0_{ER}$ and $H^0_{ER}$ both have half-lives of ≈5 h), but together they result in more than 15-fold increase in the protein's half-life.

We also estimated the half-lives of the cell surface CLIMP-63 species. $E^1_{PM}$ had an ~4 times longer predicted half-life than that of $E^0_{PM}$, consistent with the stabilizing effect of the APT2 inhibitor ML349 on surface CLIMP-63 (Fig. 3f), and its acylation-dependent association with lipid microdomains[33] (Fig. 2j, k, Supplementary Fig. 2h, i).

Altogether the model and its validation show that following synthesis and folding of CLIMP-63 into trimers, these elementary units rapidly undergo S-acylation by ZDHHC6 and subsequently assemble into higher-order complexes, presumably dimers of trimers. Jointly, but not individually, S-acylation and higher-order assembly dramatically stabilize CLIMP-63, and therefore $H^2_{ER}$ becomes the most abundant CLIMP-63 species in the cell. CLIMP-63 trimers can exit the ER, but only if they are neither acylated, nor assembled into H. CLIMP-63 that does exit the ER can reach the plasma membrane where it can be acylated by ZDHHC2 and 5[33,37]. This increases the surface residence time of CLIMP-63, probably delaying its endocytosis and transport to lysosomes for degradation.

## Regulation of ER morphology by CLIMP-63 S-acylation

In addition to its role in connecting the ER to the microtubule network[20,27], CLIMP-63 has been proposed to control the structure and abundance of ER sheets[6]. Our finding that ZDHHC6 expression modulates the cellular levels and distribution of CLIMP-63 raises the possibility that this acyltransferase may also regulate ER morphology. In support of this hypothesis, both reducing or increasing ZDHHC6 expression altered the ER: a reduced perinuclear ER density was observed in ZDHHC6 KO cells (Supplementary Fig. 6a, b), whereas drastic ER rearrangements, leading to the appearance of dilated-ER structures, with apparent reduction of the reticular morphology in cells with strong overexpression of ZDHHC6 (concentrated in dot-like structures, explained below) (Fig. 6a, b and Supplementary Fig. 6c). This phenotype was dependent on CLIMP-63 acylation since it was not observed upon expression of the acylation-deficient C100A mutant in shCLIMP-63 cells (Fig. 6c, Supplementary Fig. 6c). We observed this ER-dilation phenotype upon overexpression of ZDHHC6 in various cell types, such as U2OS cells (Supplementary Fig. 6d). It appears specific to ZDHHC6 overexpression since it was not triggered by overexpression of ZDHHC2 or other unrelated, ER-localized ZDHHC enzymes (Supplementary Fig. 6e). These morphological changes are not a consequence of massive ER stress since over-expression of ZDHHC6 led only to mild eIF2A phosphorylation (as compared to tunicamycin treatment), insignificant XBP1 splicing and no change in the mRNA levels of major ER stress mediators such as Bip, Ire1, PERK and ATF6 (Supplementary Fig. 6f–h). Consistently, global protein synthesis ($^{35}$S

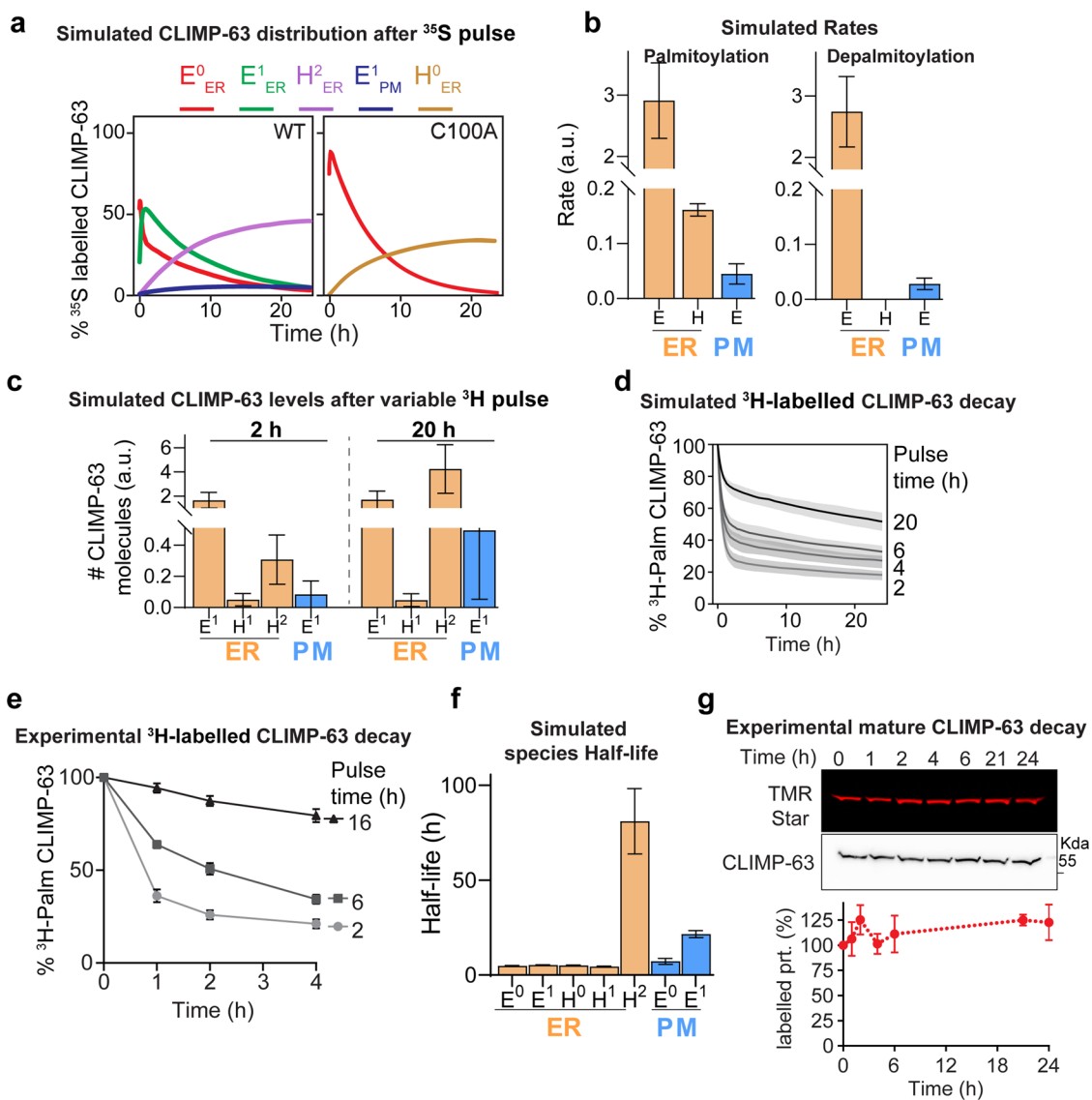

**Fig. 5 | Validation of CLIMP-63 palmitoylation modelling. a** Simulation of the levels of differently labelled CLIMP-63 species throughout time during a 35S-Cys/Met pulse-chase experiment. **b** Predicted palmitoylation and depalmitoylation rates for different CLIMP-63 species (ER-yellow, PM-blue). Rate for H species correspond to its sum in the ER (H0 + H1for Palmitoylation and H2 + H1 for Depalmitoylation). **c**, **d** Simulation of the levels of labelled CLIMP-63 after the indicated 3H-palmitate pulse period. **d** Evolution of labelled CLIMP-63 after the indicated H-palmitate pulse periods. **e** Experimental pulse-chase experiment after either 2, 6, or 16 h of metabolic labelling with 3H-palmitate. Values were normalized to the initial population (t0) as 100%. **f** Simulated bona fide half-lives of the different CLIMP-63 species. **g** Fluorescent and Western blot analysis of the decay of SNAP-CLIMP-63 in HeLa shCLIMP-63 cells labelled with TMR-star, and chased for the indicated time points. Values were normalized to t0 as 100%. For **e** and **g**, results are mean ± SD, n = 3 biologically independent experiments. Simulation data sets represent the median, and error bars the first and third quartile or SD (bar charts) through the simulation of n = 100 models. further details of the in silico labelling experiments can be found in the supplementary information−supplementary methods section.

Cyst/Met metabolic labelling) was not reduced, and even slightly increased by over-expression of ZDHHC6 but not its catalytic inactive variant[40] (Supplementary Fig. 6i).

To confirm the importance of CLIMP-63 acylation in the control of ER morphology, we searched for a means to accelerate the formation of acylated higher order complexes (H2ER). Our mathematical model suggested that one way to achieve this would be to slow down de-acylation (Fig. 6d). Accelerated formation of H2ER (Fig. 6d) would also lead to slow down CLIMP-63 decay (Fig. 6d). We have previously observed that the presence of multiple neighbouring cysteines, such as the dual acylation of calnexin in the vicinity of the transmembrane domain, slows down deacylation[45]. By analogy, we introduced a second cysteine adjacent to Cys-100, generating CLIMP-63-CC. This cysteine insertion is unlikely to have structural consequences since the cytosolic tail of CLIMP-63 is predicted to be disordered (https://iupred2a.elte.hu/). CLIMP-63-CC

was properly expressed in cells and showed a Blue NATIVE profile equivalent to WT and C100A CLIMP-63, a result that further confirms that S-acylation has not obvious effect on higher-order CLIMP-63 assembly (Supplementary Fig. 6j).

We next tested the prediction that CLIMP-63-CC would deacylated slower and have a longer half-life. 3H-palmitate pulse-chase experiments demonstrated that the rate of depalmitoylation of CLIMP-63-CC was drastically slower than that of WT, with an almost fivefold increase in the apparent half-life of bound palmitate (Fig. 6e). Metabolic 35S Cys/Met labelling experiments showed that CLIMP-63-CC was also more stable than WT CLIMP-63 (Fig. 6f). Also consistent with increased acylation in the ER, the presence of CLIMP-63-CC at the plasma membrane was lower than WT, and undetectable in DRMs (Fig. 6g, h). These experiments indicate that CLIMP-63-CC had

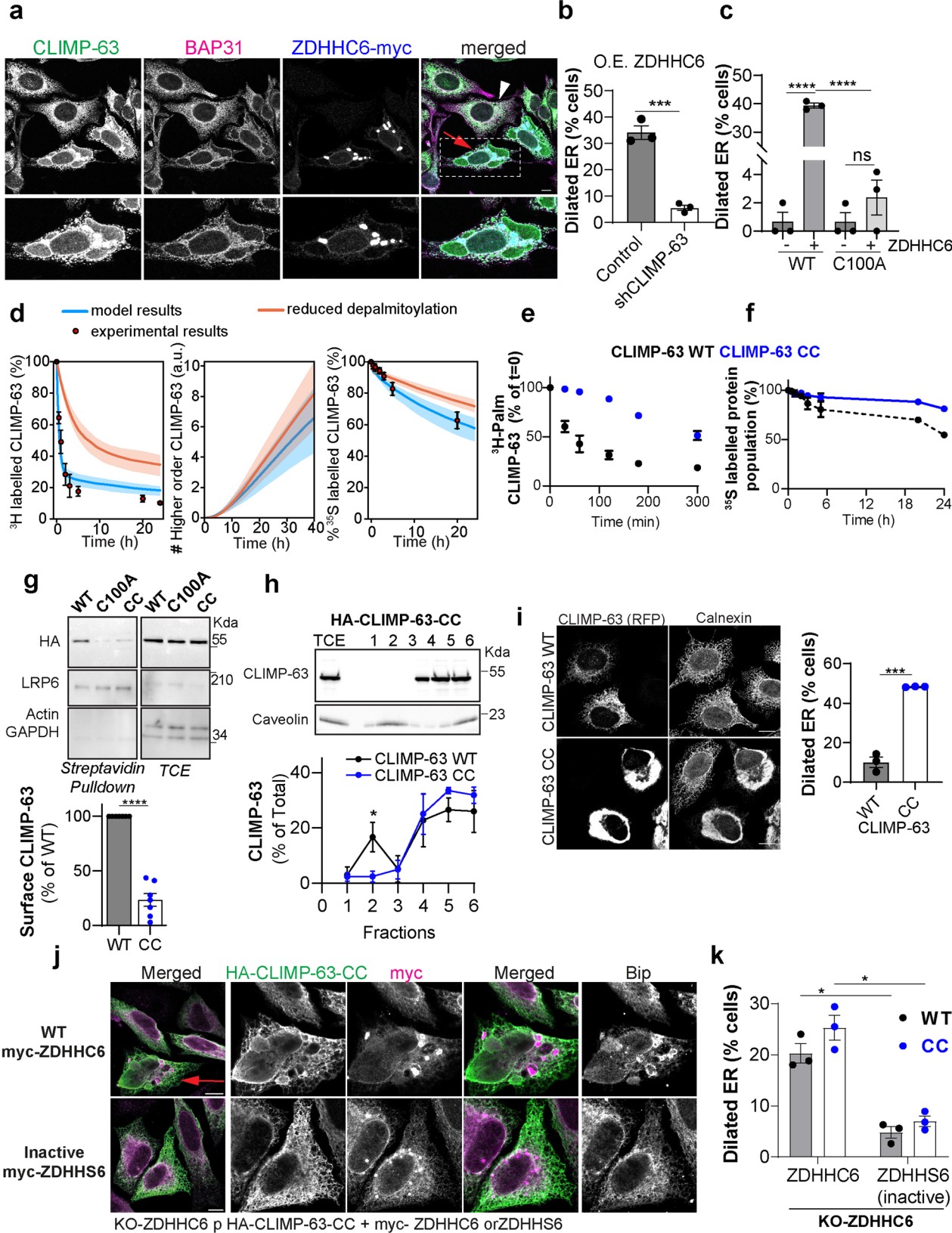

reduced ER depalmitoylation, which in turn increases its ER retention, diminishing its surface expression.

We next evaluated the consequences of CLIMP-63-CC on ER morphology. Confocal analysis of shCLIMP-63 cells overexpressing CLIMP-63-CC showed a striking densification of perinuclear ER-sheets (Fig. 6i) and the number of cells with dilated ER was strongly increased

when compared to those expressing WT CLIMP-63 (Fig. 6i). The CLIMP-63-CC induced ER dilation was dependent on ZDHHC6 catalytic activity (Fig. 6j, k). Imaging by super resolution, structured illumination microscopy (SIM) showed that the ER was however still structured but with increased un-fenestrated sheet-like regions (Supplementary Fig. 6k, middle panels).

**Fig. 6 | ZDHHC6-mediated CLIMP-63 S-acylation controls ER morphology.**
**a** Confocal images of HeLa cells expressing ZDHHC6-myc immunolabelled for myc (blue), BAP31 (magenta), and CLIMP-63 (green). Red arrow and inset show dilated ER in ZDHHC6-myc expressing cells. White arrowhead shows bystander cell.
**b** Quantification of the percentage of ZDHHC6-myc expressing cells with ER dilation in control or shCLIMP-63 HeLa cells. Results are mean ± SEM ($n = 3$ counting total: Control, 129 cells; shCLIMP-63, 226 cells; ***$p = 0.0005$, obtained by unpaired, two-tailed student's $t$ test. **c** Same as in **b** in shCLIMP-63 cells co-overexpressing or not myc-ZDHHC6 with HA-CLIMP-63 WT or C100A. Results are mean ± SEM ($n = 3$) counting total: HA-CLIMP-63-WT, 137 cells, HA-CLIMP-63-WT + ZDHHC6, 79 cells, HA-CLIMP-63-C100A, 145 cells, HA-CLIMP-63-C100A + ZDHHC6, 182 cells (****$p < 0.0001$, obtained by one-way ANOVA, Tukey's multiple comparison).
**d** Computational simulation of CLIMP-63 depalmitoylation (left), Higher-order assembly (middle), and protein stability (right) upon normal (blue) and slower (orange) depalmitoylation kinetics. Median shown by solid lines, 1st and 3rd quartile by shaded interval. **e, f** Quantification of CLIMP-63. **e** $^3$H-palmitate decay or **f** apparent decay in shCLIMP-63 cells expressing HA-CLIMP-63 WT or CC, pulsed with $^3$H-palmitate pulse (2 h) or $^{35}$S metabolic (20 min) and followed by the indicated chase period. Results set to 100% for $T = 0$ min are mean ± SD, $n = 3$.
**g** Western blots of surface biotinylated proteins and total cell extracts (TCE) from shCLIMP-63 cells expressing HA-CLIMP-63 WT, C100A or CC mutant. LRP6 and

actin/GAPDH are positive and negative controls, respectively. Surface CLIMP-63 results normalised to WT are mean ± SEM ($n = 7$), (****$p < 0.0001$, obtained by unpaired, two-tailed student's $t$ test.). **h** Western blot analysis of fractionated cell lysates from cells transfected as in **e** (DRMs in fraction 2 are marked by caveolin). HA-CLIMP-63-CC in each fraction was compared to WT HA-CLIMP-63 levels obtained in parallel experiments depicted in Supplementary Fig. 2h. Results are mean ± SEM ($n = 3$), (*$p = 0.0255$, obtained by two-way ANOVA, Sydak's multiple comparison). **i** Confocal images and quantification of the percentage of cells with ER dilation in shCLIMP-63 HeLa cells transfected with RFP-CLIMP-63 WT or CC, immunolabelled for calnexin. Results are mean ± SEM ($n = 3$), WT: 91 cells, CC: 60 cells (***$p = 0.0001$, obtained by unpaired, two-tailed student's $t$ test.). **j, k** Airyscan-confocal images and quantification of the percentage of cells with dilated ER in KO-ZDHHC6 cells transfected with HA-CLIMP-63 WT or CC plus myc- WT-ZDHHC6 or inactive mutant ZDHHS6 immunolabelled for HA, myc and ER marker Bip. Results are mean ± SEM ($n = 3$) with >200 cells counted per condition (*$p = 0.0179$ and *$p = 0.0179$ obtained by two-way ANOVA, Sydak's multiple comparison). Red arrow and inset show dilated ER in ZDHHC6-myc expressing cells. Unless otherwise indicated all means were derived from biologically independent experiments. Simulated data was derived from the simulation of $n = 100$ models. further details of the in silico labelling experiments can be found in the supplementary information – supplementary methods section. All scale bars: 10 μm.

Overexpression of acylation deficient C100A CLIMP-63 led to alterations of the ER reticular network (Supplementary Fig. 6k) in agreement with the recently proposed role for S-acylated CLIMP-63 in organizing ER-mitochondrial contact sites[47]. Altogether, these observations show that altering the dynamics of acylation and deacylation of CLIMP-63 influence the morphology of the ER.

### CLIMP-63 S-acylation controls fenestration of ER sheets
To gain further insight into the changes in ER-architecture caused by CLIMP-63 acylation by ZDHHC6, we performed correlative electron microscopy (EM). The expression of RFP-tagged variants of CLIMP-63 enable the identification of transfected cells (Fig. 7a). Expectedly, shCLIMP-63 cells expressing WT CLIMP-63 displayed well-organised ER-sheets whereas cells expressing the acylation deficient mutant presented a general decreased ER density, as well as a disorganisation of the ER network (Fig. 7a), as observed by SIM (Supplementary Fig. 6h). Expression of CLIMP-63-CC led to a strong densification of ER sheet-like compartments (Fig. 7a), in agreement with the confocal microscopy analysis (Fig. 6i–k). A similar ER densification was observed in cells expressing WT RFP-CLIMP-63 under conditions of ZDHHC6-myc overexpression (Fig. 7a). High ZDHHC6 expressing cells could clearly be identified by the presence of bright ER clusters (Supplementary Fig. 7a), as also detected in the confocal microscopy analyses (Fig. 6a, j). These clusters represent highly organised ER structures, known as OSERs (Organised Smooth ER)[48] (Supplementary Fig. 7), which are distinct from ER stress-induced ER whorls[49].

We next performed focused ion beam scanning electron microscopy (FIBSEM). This technique provides serial images with near isotropic voxels from which a reconstruction of an ER volume can be generated (Fig. 7b). In control conditions, i.e., endogenous ZDHHC6 expression, the ER sheets formed a stratified matrix with multiple clustered and complex fenestrations between layers. In cells with high ZDHHC6 expression, i.e. containing ZDHHC6-induced OSERs, the pattern of ER sheet layers was strikingly denser, with reduced fenestrations, and abundant membrane convolutions (Fig. 7b). The FIBSEM images and their 3D reconstruction confirmed that overexpression of ZDHHC6 strongly increased continuity and densification of the ER sheets.

Quantifying alterations of the ER morphology remains a major challenge for cell biology image analysis. To accurately measure the ER densification phenotype induced by ZDHHC6, we employed persistent homology, a mathematical tool in applied algebraic topology (for background and mathematical introductions please refer to the supplementary methods and previous studies)[50–52]. Persistent homology

tracks the appearance or disappearance of features—such as spherical cavities (in degree-2) and loops (in degree-1)—in data-sets across a range of distance scales (Supplementary Fig. 8). Data is shown as a persistence diagram, which tracks all membrane features throughout the ER-3D reconstructions analysed. Each point refers to a feature, where the horizontal coordinate encodes its appearance, and the vertical, the disappearance. Therefore, abundance in small and noisier features (e.g. resulting from small fenestrations, nanoholes) will correspond to values closer to the diagonal of the diagram, whereas larger, more significant features (e.g. expanded membrane sheets) will have higher persistence values and be farther from the diagonal (Fig. 7c, d and Supplementary Fig. 8). Persistent homology analysis, particularly in degree-2, confirmed the prominent change in ER topology caused by ZDHHC6 overexpression, which promotes the expansion of large and dense ER-sheets and reduces the amount and complexity of ER fenestrations (Fig. 7c, d).

## Discussion
CLIMP-63 is an enigmatic protein, about which there are many open questions. Here we addressed the impact of S-acylation and its dynamics. We used a variety of experimental approaches − biochemistry, microscopy, metabolic labelling− to describe some behavioural aspects of CLIMP-63, and mathematical modelling to understand their complexity and interconnectedness. Altogether the work led us to propose the following scenario. CLIMP-63, synthesized by ribosomes on the ER membrane, is co-translationally inserted into the membrane with its large C-terminal domain in the lumen, where it rapidly folds and assembles, in *cis*, into trimeric elementary units ($E^0_{ER}$) (Fig. 4b). The lack of classical ER retention signals within CLIMP-63 allows a minor population of folded $E^0_{ER}$ to exit the ER and reach the plasma membrane. The majority, however, is retained in the ER through two independent mechanisms: S-acylation on a single cysteine and higher-order assembly, most likely dimers of trimers ($H_{ER}$). $E^0_{ER}$ are highly susceptible to S-acylation by ZDHHC6, rapidly generating acylated trimers. $E^1_{ER}$ can promptly be de-acylated by APT2, in a cycle that sustains limited exit from the ER. The majority of $E^1_{ER}$ however, assembles into S-acylated complexes ($H^2_{ER}$). This $H^2_{ER}$ form is protected from de-acylation and has ~16-times longer half-life than any other ER-localized CLIMP-63 species, making it the major species in the cell at steady state. In the absence of S-acylation, higher-order assembly can still occur, retaining CLIMP-63 in the ER. $H^0_{ER}$ is however less stable and less abundant. The non-acylated elementary units $E^0_{ER}$ that exit the ER are substrates for two other acyltransferases that localize to the late secretory-endosomal system, ZDHHC 2 and 5[33,37,38].

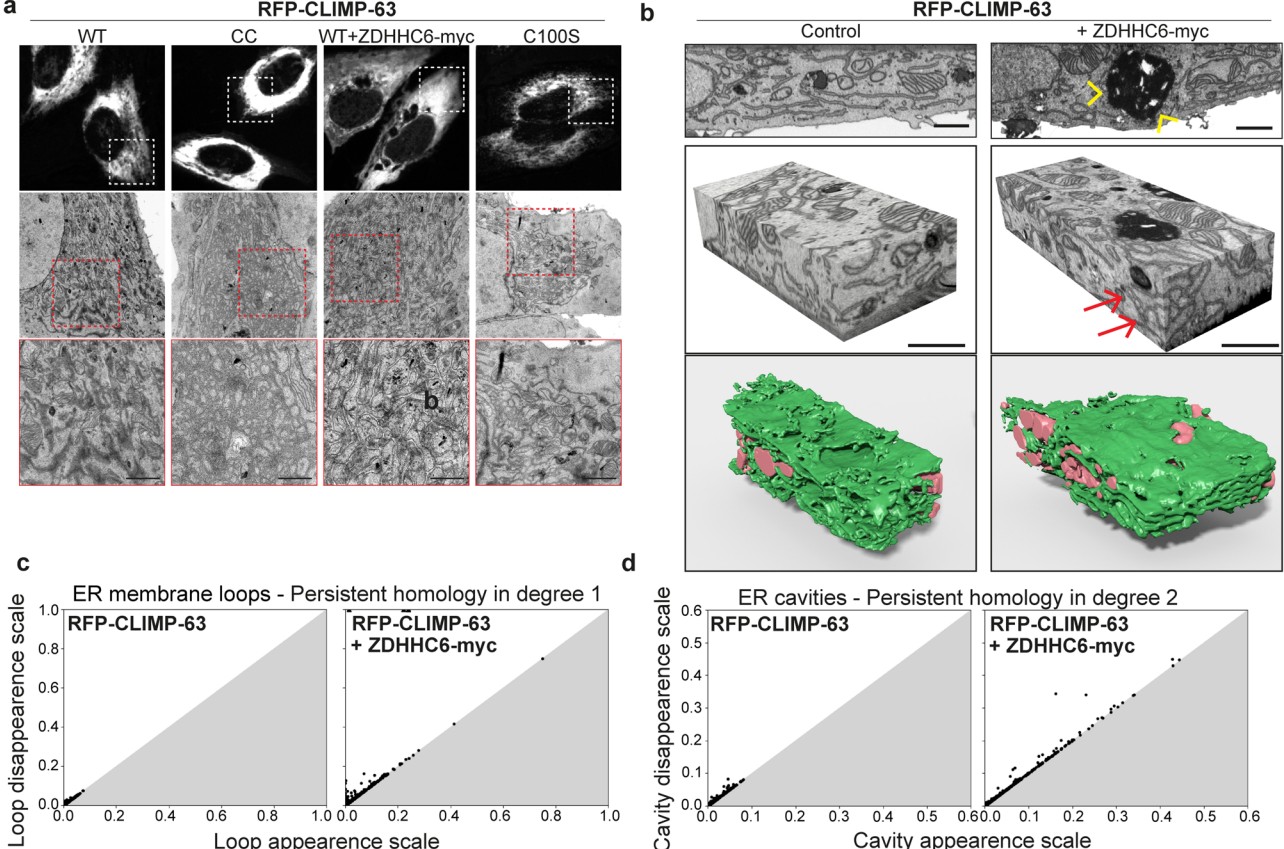

**Fig. 7 | Ultrastructure analysis of ER morphology. a** Correlated light and electron microscopy of shCLIMP-63 HeLa cells overexpressing RFP-CLIMP-63 WT, CC or C100S, or RFP-CLIMP-63 WT together with ZDHHC6-myc. Light microscopy images (top) with the boxed region (red) indicating the area imaged with TEM (middle) and zoomed region (bottom) (second red box). Scale bars: 1 μm. **b** FIBSEM was used to 3D image the ER in RFP-CLIMP-63 in control or upon overexpression of ZDHHC6 (detected by the presence of OSERs−yellow arrowheads). FIBSEM image stacks depict the convoluted branching pattern of the ER. Numerous closed loops of ER membrane can be seen in the two imaging planes upon ZDHHC6-myc expression (red arrows). Reconstruction of ER (green) with the reconstructed mitochondria (pink). Scale bars: 1 μm. **c** Quantification of ER-membrane loops by degree-1 persistent homology and **d** ER cavities by degree-2 persistent homology in control and ZDHHC6-myc overexpression.

S-acylation is essential for the maintenance of CLIMP-63 at the cell surface, within raft-like nanodomains[33] controlling its residence time there and thus influencing its signalling capacity.

The acylation of CLIMP-63 not only affects its intracellular trafficking and cellular stability, but also its ability to shape the ER. We analysed the ER morphology under conditions where the CLIMP-63 acylation-deacylation kinetics were modified, i.e., accelerated acylation by ZDHHC6 overexpression, delayed deacylation using the double cysteine CC mutant or no acylation by mutating Cys-100. All CLIMP-63 variants formed similar higher-order structures (Supplementary Fig. 6j), yet their effects on ER morphology were different indicating that adequate acylation levels and dynamics of CLIMP-63 are necessary for adapted morphology. When acylation was excessive, we observed a loss of fenestration and a massive expansion of ER sheets. These findings are consistent with a recent studie using live-cell stimulated depletion (STED) microscopy showing that CLIMP-63 coordinates the formation of dynamic nanoholes within ER sheets and luminal ER nanodomain heterogeneity[30,31]. The present work suggests that the control of nanohole formation is tuned through the acylation of CLIMP-63. The addition of medium chain fatty acids to CLIMP-63 trimers and higher-order structures is likely to modify the lipid composition and/or physical-chemical properties of the surrounding membranes, and possibly thereby the intrinsic membrane curvature. A recent computational analysis indeed proposes that membrane tension and curvature[25], both of which could well be influence by CLIMP-63 acylation and lateral lipid organisation, are the key elements that drive nanohole formation.

How CLIMP-63 acylation cycles are controlled remains to be established. The metabolic state of cells and tissues is likely to play a role. It was indeed recently observed that the ER organization was disrupted in hepatocytes from obese mice, due to an imbalance between the levels of CLIMP-63 and ER tubule-associated proteins, which could be rescued by the exogenous overexpression of CLIMP-63[53]. Another recent study found that excess fatty acid synthesis leads to the densification of ER membranes causing downstream mitotic complications[54]. A link between lipid metabolism and protein acylation, although expected, remains to be explored and mechanistically understood. Future studies should also address the structural features that enable acylated CLIMP-63 to control ER fenestration, whether this property cross-talks to its microtubule binding ability, and finally, the exact mechanism by which the still mysterious CLIMP-63 luminal domain influences ER sheet formation.

## Methods

### Plasmids and antibodies

For western blotting and immunofluorescence, myc (RRID:AB_2537024) and Bap31 (RRID:AB_325095) antibodies were from Thermo Fisher (US). Anti-CLIMP-63 were either from Alexis/ENZO (G1/296, CH, RRID:AB_2051140) or Bethyl Laboratories (A302, RRID:AB_1731083). anti-LRP6 were also from Bethyl Laboratories (US), (RRID:AB_21393299). Anti-calreticulin (RRID:AB_1267911), anti-Spastin

(RRID:AB_2042945) and anti-BiP (RRID:AB_880312) were from Abcam (UK). Anti-tubulin (RRID:AB_477579), anti-GAPDH (RRID:AB_2533438), anti-ZDHHC6 (RRID:AB_2304658), anti-FLAG (RRID:AB_439685), anti-LPXN (RRID:AB_1853250), anti-Caveolin1(RRID:AB_476842) and anti-transferrin receptor (RRID:AB_86623) were from Sigma (US). Anti-actin was from Millipore (US) (RRID:AB_2223041). Anti-HA was from BioLegend (US) (RRID:AB_2563418). Anti-GFP (RRID:AB_2336883) and anti-RFP (RRID:AB_ 2336063) were from Roche (CH). Anti-calnexin was previously described 1 and provided by Dr. M. Molinari. Anti-TRAPα was provided by Dr. R. Hegde. Anti-HA-HRP conjugated was from Roche (CH) (RRID:AB_390918). Rabbit anti-EIF2alpha (Cell Signalling #9722), rabbit anti-eIF2 alpha (Phospho-Ser51) (Biorbyt #orb5998, RRID:AB_10928244). For immunoprecipitation, sepharose G-beads were from GE Healthcare (US), anti-myc-beads were from Thermo Fisher (US) and anti-HA-beads were from Roche (CH).

The siRNAs for ZDHHC2 (TAGCTACTGCTAGAAGTCTTA), ZDHHC3 (TCCGTTCTCATGAATGTTTAA), ZDHHC5 (ACCACCATTGCC AGACTACAA) and ZDHHC6 (GAGGTTTACGATACTGGTTAT) were from Qiagen, D. As control siRNA, we used either the AllStars negative control siRNA (Qiagen, D) or targeted the viral glycoprotein VSV-G (sequence: ATTGAACAAACGAAACAAGGA).

Point mutations were generated using QuikChange II XL kit from Agilent Tech (US). ZDHHC6-GFP was obtained by inserting the PCR amplified product of ZDHHC6 in a peGFP-C3 vector using XhoI and BamHI sites. CLIMP-63-HA was generated by inserting CLIMP-63-HA cDNA in place of the RFP in a pTagRFP vector. The following constructs were kind gifts: CLIMP-63-YFP from Dr. Hans-Peter Hauri and Dr. Hesso Farhan; ZDHHC2-myc, ZDHHC6-myc and ZDHHC16-FLAG from Dr. Masaki Fukata.

## Cell culture, transfections and drug treatments
All HeLa cells were cultured in MEM Eagle (Sigma, US) complemented with 10% FCS (PAN Biotech, D), 1% Pen/Strep, 1% L-Glutamine, and 1% MEM NEAA (all Gibco, US). They were mycoplasma negative as tested on a trimestral basis using the MycoProbe Mycoplasma Detection Kit CUL001B. RPE-1 cells were grown in complete Dulbeccos MEM (DMEM, Sigma) at 37 °C supplemented with 10% foetal bovine serum (FBS), 2 mM L-Glutamine, penicillin and streptomycin. For transfection, cells were dissociated using trypsin and plated in tissue culture dishes (Falcon, US). After 24 h, the medium was changed and the cells were transfected using Fugene for plasmids (Promega, US) or INTERFERin (Polyplus, F) for silencing with siRNA. The cells were incubated for 24 h to 48 h (for plasmids) or 72 h (for siRNA) before performing experiments. Drug treatments were used at: nocodazole (2 h at 10 μg/mL), Taxol (4 h at 5 μg/mL) both in IM medium (described in ³H-labelling). Taxol treatments for IF were done in complete medium. ML348 and ML349 were used at 10 μM in complete medium for 4 h of pre-treatment followed by the indicated time before harvest. Tunicamycin was used at 10 μg/ml for 4 h in complete medium.

## shRNA stable cell lines and qPCR
The stable HeLa cell lines transduced with shRNA were generated as described elsewhere[55]. To summarize, the shRNA of interest was inserted in a pRRLsincPPT-hPGK-mcs-WPRE vector. HEK293T cells were co-transfected with pMD2g and pSPAX2, which encode the envelope and packaging proteins, respectively. Lentiviral particles were harvested and titrated by qPCR. Finally, low passage HeLa cells were transduced with a range of viral loads and tested by qPCR and by Western blot to quantify the silencing efficiency of the targeted protein. Cells were maintained in 8 μg/mL puromycin. The sh-control consisted of the parent vector with non-targeting sequence. The shRNA sequence against ZDHHC6 was GATCcccCCTAGTGCCATG ATTTAAAttcaagagaTTTAAATCATGGCACTAGGttttttC and against CLIMP-63 was GATCcccGAGGTAACTATGCAAAGCAttcaagagaTGCTTT GCATAGTTACCTCttttttC. Real-time quantitative PCR was performed as described previously[39]. Additional qPCR primers are described in Supplementary Information.

## CRISPR/Cas9 KO of ZDHHC6
CRISPR/Cas9 KO of ZDHHC6 was obtained following previously published protocols[56] using the following guide RNA sequence targeting exon 2 of ZDHHC6: TGGGGTCCCATCATAGCCCT. Cells were selected using 10 μg/ml of puromycin and blasticidin.

## Immunoprecipitation and Western blotting
For immunoprecipitation and Western blot, cells were lysed on ice for 30 min with lysis buffer (500 mM Tris–HCl pH 7.4, 2 mM benzamidine, 10 mM NaF, 20 mM EDTA, 0.5% NP40 and a protease inhibitor cocktail (Roche, CH)). The lysate was then clarified by centrifugation at 4 °C for 3 min at 5000 rpm. Lysates were pre-cleared using Sepharose G-beads only for 30 min at 4 °C before immunoprecipitation (G-beads plus antibody) turning on a wheel overnight at 4 °C. The beads were then washed 3× with lysis buffer before adding 4× Sample Buffer including beta-mercaptoethanol. The samples were boiled 5 min at 95 °C and vortexed before loading and migrating on 4–12% or 4–20% Tris-glycine SDS-PAGE gels. Blots were revealed using a Fusion Solo (Vilber Lourmat, CH) and quantified with ImageJ or Bio1D (Vilber Lourmat, CH).

## Acyl-RAC
Acyl-RAC was performed according to ref. [57]. In brief, a post nuclear supernatant was retrieved and the proteins were blocked in a buffer with 0.5% TX100, a protease inhibitor cocktail and 1.5% MMTS for 4 h at 40 °C vortexing every 15 min. The proteins were then precipitated using cold acetone at −20 °C for 20 min and centrifuged at 4 °C for 10 min at 7500 rpm. The pellet was washed 5× with 70% acetone. After drying, the samples were resuspended in an SDS buffer. 10% of the sample was reserved as input and the rest was separated into two tubes. The first tube was treated with hydroxylamine 0.5 M (final, in Tris pH 7.4) and 10% thiopropyl sepharose beads (Sigma). The second tube (negative control) had only Tris-HCl pH 7.4 with 10% thiopropyl sepharose beads. The samples were incubated at RT overnight. Finally, the beads were washed 3× in SDS-buffer, before adding sample buffer (4×) w/beta-mercaptoethanol and performing SDS-PAGE followed by a western blot as described above.

## APEGS (PEGylation)
The stoichiometry of protein S-Palmitoylation was assessed by APEG. The assay was followed as described elsewhere[58], with minor modifications. Hela cells were lysed in 4% SDS, 5 mM EDTA, in PBS with complete Protease Inhibitor Cocktail (Roche). Supernatant proteins were retrieved after centrifugation at $100,000 \times g$ for 15 min. The proteins were reduced with 25 mM TCEP for 1 h at 25 °C, and free cysteine residues were blocked with 20 mM NEM for 3 h at 25 °C. After chloroform/methanol precipitation, the proteins were resuspended in PBS with 4% SDS and 5 mM EDTA and incubated in 1% SDS, 5 mM EDTA, 1 M NH₂OH, pH 7.0 for 1 h at 37 °C. As a negative control, 1 M Tris-HCl, pH 7.0, was used. After precipitation, the proteins were resuspended in PBS with 4% SDS and PEGylated with 20 mM mPEGs for 1 h at 25 °C to label newly exposed cysteinyl thiols. As a negative control, 20 mM NEM was used instead of mPEG (5 kDa-PEG). After precipitation, proteins were resuspended in SDS-sample buffer and boiled at 95 °C for 5 min. The proteins were separated by SDS-PAGE, transferred and western blotted. Protein concentration was measured by BCA protein assay.

## Isolation of detergent-resistant membranes (DRMs)
Approximately $1 \times 10^7$ cells were re-suspended in 0.5 ml cold TNE buffer (25 mMTris-HCl, pH 7.5, 150 mM NaCl, 5 mM EDTA, and 1% Triton X-100) with a tablet of protease inhibitors (Roche). Membranes were solubilized in a rotating wheel at 4 °C for 30 min. DRMs were isolated using an Optiprep™ gradient: the cell lysate was adjusted to

40% Optiprep™, loaded at the bottom of a TLS.55 Beckman tube, overlaid with 600 μl of 30% Optiprep™ and 600 μl of TNE, and centrifuged for 2 h at 55,000 rpm at 4 °C for cells. Six fractions of 400 μl were collected from top to bottom. DRMs were found in fractions 1 and 2. Equal volumes from each fraction were analysed by SDS-PAGE and western blot analysis using anti-CLIMP-63, HRP-conjugated anti-HA, caveolin1 and transferrin receptor antibodies.

### Surface biotinylation
Cells were allowed to cool down shaking at 4 °C for 15 min to arrest endocytosis. Cells were then washed three times with cold PBS and treated with EZ-Link Sulfo-NHS-SS-Biotin No weight for 30 min shaking at 4 °C. Cells were then washed three times for 5 min with 100 mM NH4Cl and lysed in 1% Tx-100 to do DRMs or in IP Buffer for 1 h at 4 °C. Lysate were then centrifuged for 5 min at 5000 rpm and the supernatant incubated with streptavidin agarose beads overnight on a wheel at 4 °C. Beads were washed with IP buffer 5 times and the proteins were eluted from the beads by incubation in SDS sample buffer with ß-mercaptoethanol for 5 min at 95° buffer prior to performing SDS-PAGE and western blotting.

### ³H-metabolic labelling
Cells were seeded in tissue culture dishes as described above. For labelling, the cells were starved using IM medium (Glasgow minimal essential medium buffered with 10 mM Hepes, pH 7.4). After 1 h, the medium was replaced by IM with 3H-palmitate at 200 μCi/mL (American Radiolabeled Chemicals, US) for 2 h at 37 °C. Cell lysis, immunoprecipitation and SDS-PAGE were performed as above. The gels were fixed for 30 min with 10% acetic acid, 25% isopropanol in water and the signal was amplified for 30 min with NAMP100 (GE Healthcare, US). The gels were then dried and applied to an Amersham Hyperfilm MP (GE Healthcare, US). The radioactivity was visualized and quantify using a Typhoon TRIO (GE Healthcare, US).

### ³⁵S Pulse chase metabolic labelling
The cells were plated in tissue culture dishes as described above. 48 h post-transfection, the cells were starved 30 min at 37 °C in DMEM-HG medium (devoid of Cys/Met). The pulse consisted of 70 μCi/mL 35 S (American Radiolabeled Chemicals, US) in the same starvation medium for 20 min at 37 °C. Cells were then washed 2× and incubated in complete MEM medium containing Cys/Met in excess. Finally, cells were lysed and harvested. Proteins of interest were immunoprecipitated and prepared for western blotting as previously described.

### Protein production and purification
Suspension-adapted HEK293E cells transiently transfected with construct expressing CLIMP-63 Luminal Domain with N-terminal signal recognition peptide and either N- or C-terminal His6-FLAG tag using PEI MAX (Polysciences) in RPMI-1640 (Gibco) supplemented with 0.1% Pluronic-F68. After 1.5 h, cells were diluted into Excell293 medium (Sigma) supplemented with 4 mM glutamine and 3.75 mM valproic acid and agitated for 37 °C. Following a 7-day incubation the cell culture medium was harvested by centrifugation and clarified using a 0.22 μm filter. The conditioned medium was purified by Ni-NTA affinity chromatography via CLIMP63's His-tag followed by gel-filtration chromatography in 500 mM NaCl, 50 mM HEPES pH 7.5.

### Intact protein mass LC-MS analysis under native-like conditions
To preserve non-covalent interactions, intact mass measurements were performed under native-like conditions by injecting the samples into MAbPac SEC-1 column (300 Å, 5 μm, 4 × 150 mm, Thermo Fisher Scientific, Sunnyvale, CA, USA) using a Dionex Ultimate 3000 analytical RSLC system (Dionex, Germering, Germany) coupled to a HESI source (Thermo Fisher Scientific, Bremen, Germany). The isocratic separation was performed within 7 min at flow rate of 300 μl/min and 50 mM ammonium acetate, pH 7.5 as mobile phase. Eluting fractions were analysed on high resolution QExactive HF-HT-Orbitrap-FT-MS benchtop instrument (Thermo Fisher Scientific, Bremen, Germany). High-mass-range (HMR) mode was activated with resolution of 15,000, in-source CID of 50 eV and AGC (automatic gain control) target of 5e6. The scan range was set to 1900–8000 $m/z$. Data analysis was performed with Protein Deconvolution 4.0 (Thermo Fischer Scientific, Sunnyvale, CA, USA) using Respect algorithm.

### Intact protein mass LC-MS analysis under denaturing conditions
To assess the mass of the monomeric form, intact mass measurements were performed under denaturing conditions by injecting the samples into Acquity UPLC Protein column BEH C4 (300 Å, 1.7 μm, 1 × 150 mm, Waters, Milford, MA, U.S.A.) using a Dionex Ultimate 3000 analytical RSLC system (Dionex, Germering, Germany) coupled to a HESI source (Thermo Fisher Scientific, Bremen, Germany). The separation was performed with a flow rate of 90 μl/min by applying a gradient of solvent B from 15 to 20 % in 2 min, then from 20 to 45 % within 10 min, followed by column washing and re-equilibration steps. Solvent A was composed of MilliQ water with 0.1% formic acid, while solvent B consisted of acetonitrile with 0.1% formic acid. Eluting fractions were analysed on high resolution QExactive HF-HT-Orbitrap-FT-MS benchtop instrument (Thermo Fisher Scientific, Bremen, Germany). Protein mode was activated with resolution of 15 000, in-source CID of 25 eV, AGC target of 3e6 and averaging 10 μscans. The scan range was set to 600–2000 $m/z$. Data analysis was performed with Protein Deconvolution 4.0 (Thermo Fischer Scientific, Sunnyvale, CA, USA) using Respect algorithm.

### Shotgun bottom-up proteomic analysis
Protein content of the samples was further verified with shotgun/bottom-up proteomic LC-MS/MS analysis. 5 μg of protein in 25 mM ammonium bicarbonate buffer (pH 7.8) were boiled at 95 °C for 2 min, reduced with TCEP solution of 5 mM final concentration at 55 °C for 30 min, followed by alkylation with IAA solution of 5 mM final concentration in the dark for 30 min at room temperature and digestion with trypsin (enzyme/protein ratio of 1:30 w/w) at 37 °C overnight. Reaction was quenched by acidification using formic acid to a final acid concentration of 0.1%. mObtained proteolytic peptide mixture was separated on column ZORBAX Eclipse Plus C18 column (2.1 × 150 mm, 5 μm, Agilent, Waldbronn, Germany) using Dionex Ultimate 3000 analytical RSLC system (Dionex, Germering, Germany) coupled to a HESI source (Thermo Fisher Scientific, Bremen, Germany. The separation was performed with flow rate of 250 μl/min by applying a gradient of solvent B from 5 to 35% within 60 min, followed by column washing and re-equilibration steps. Solvent A was composed of MilliQ water with 0.1% formic acid, while solvent B consisted of acetonitrile with 0.1% formic acid. Eluting peptides were analysed on QExactive HF-HT-Orbitrap-FT-MS benchtop instrument (Thermo Fisher Scientific, Bremen, Germany). MS1 scan was performed with 60,000 resolution, AGC of 1e6 and maximum injection time of 100 ms. MS2 scan was performed in Top10 mode with 1.6 $m/z$ isolation window, AGC of 1e5, 15,000 resolution, maximum injection time of 50 ms and averaging 2 μscans. HCD was used as fragmentation method with normalized collision energy of 27%.

Data analysis was performed using Trans-Proteomic Pipeline software (TPP, Institute for Systems Biology, Seattle Proteome Center) using Tandem pipeline with X-Tandem search engine. The cleavage specificity for trypsin was set with two allowed missed cleavages, precursor and product ion mass tolerances of 10 ppm and 0.02 Da, respectively. Cysteine carbamidomethylation and methionine oxidation were chosen as constant and variable modifications, respectively. The false discovery rate (FDR) was set to 1% with minimal peptide length of seven amino acids.

## Immunofluorescence

Cells were seeded on glass coverslips (N1.5, Marienfeld, D) for at least 24 h. Fixation and permeabilization were optimised to preserve either (i) the secretory pathway (cells were washed 3× with PBS, fixed with 3% paraformaldehyde for 20 min at 37 °C, washed 3× with PBS, quenched with 50 mM of NH4Cl for 10 min at RT, washed 3× with PBS, permeabilized with 0.1% Triton X-100 for 5 min at RT and finally washed 3× with PBS) or (ii) the cytoskeleton and ER membranes (cells were washed 3× with PBS, fixed with precooled methanol for 4 min at −20 °C, and washed 3× with PBS). In both cases, the cells were then blocked overnight at 4 °C (or 30–60 min at RT) in PBS + 0.5% BSA (GE Healthcare, US). The coverslips were incubated with primary antibody for 30–120 min at RT, washed 3× for 5 min with PBS − 0.5% BSA and incubated for 30–45 min at RT with secondary fluorescent antibodies (Alexa 488, 568 or 647, Invitrogen, US), and finally washed again 3× with PBS − 0.5% BSA prior to mounting in Mowiol or ProLong mounting medium (ThermoFisher Cat#P36934). The coverslips were imaged by confocal microscopy using a LSM710 microscope or LSM 980, Airyscan (Zeiss) with a 63× oil immersion objective (NA 1.4). Images were acquired using ZEN 2009, ZEN Blue ver. 3.4.91.

## Proximity ligation assay

The Duolink-PLA was performed according to the manufacturer's protocol (Sigma). At least 20 cells for each condition were imaged as one horizontal plane cutting the mid-height of the nucleus. For each cell, the number of PLA dots was counted manually and normalized to the endoplasmic reticulum area.

## Structured illumination microscopy

Cells seeded on glass cover slips (Source? 170 ± 5 μm thickness and between 18 mm and 24 mm in diameter) were processed as for immunofluorescence. Coverslips were imaged using an inverted Nikon Eclipse Ti Motorized microscope, with Andor iXon3 897 detector using a APO TIRF 100x (NA 1.49) oil immersion objective (working distance of 0.12 mm). Imajes were acquired using NIS Elements with JOBS.

## Correlative electron microscopy

Cells were plated and transfected with ZDHHC6-GFP plasmids on glass coverslips coated with a 5-nm layer of carbon outlining a numbered grid reference pattern. After 24 h, the cells were fixed for 60 min in a buffered solution of 2% paraformaldehyde and 2.5% glutaraldehyde at 25 °C, and then washed 3× with cacodylate buffer. The coverslips were then mounted in a holder for fluorescence microscopy and the cells imaged by confocal microscopy (LSM700, Zeiss, 63× objective, NA 1.4). The cells of interest were imaged at a range of magnifications and their location recorded according to the carbon grid pattern. The coverslips were then post-fixed with 1% osmium tetroxide and 1.5% potassium ferrocyanide in cacodylate buffer (0.1 M, pH 7.4) for 40 min at 25 °C. After washing in distilled water and further staining with osmium alone followed by 1% uranyl acetate, they were dehydrated in a series of increasing concentrations of alcohol, then embedded in Durcupan resin, which was hardened overnight at 65 °C. The next day, the resin containing the cells of interest was separated from the coverslips and mounted onto a blank resin block for ultrathin sectioning. Serial ultrathin sections were cut at 50 nm thickness and collected onto a formvar support film on single slot copper grids. Images were acquired at 80 kV using a transmission electron microscope (Tecnai Spirit, FEI Company, US).

## Focused Ion beam scanning electron microscopy (FIBSEM)

Cells of interest, recorded with fluorescent microscopy and prepared for electron microscopy (see above), were serially imaged using FIB-SEM. Resin blocks were trimmed using an ultramicrotome so that the cell was located within 5 μm of the edge. This block was then glued to aluminium stub, coated with a 20-nm layer of gold in a plasma coater, and placed inside the microscope (Zeiss NVision 40, Zeiss NTS). An ion beam of 1.3 nAmps was used to sequentially mill away 10-nm layers of resin from the block surface to enable the cell to be serially imaged. Images were collected using the backscatter detector with the electron beam at 1.6 kV and grid tension set at 1.3 kV to collect only the highest energy electrons.

The final images were precisely aligned using the StackReg algorithm[56] in ImageJ, and the ER, mitochondria, nuclear membrane, and cell membrane were segmented using the Microscopy Image Browser software[57]. The mesh models were then exported to the Blender software (www.blender.org) for final rendering and visualization.

## Statistics and reproducibility

Statistical analyses were carried using Prism software. Data representation and statistical details can be found in the figure legends. Unless otherwise indicated, an unpaired two-tailed Student's $t$-test was used for direct comparison of means between two groups, whereas ANOVA was used to compare the means among three or more groups. For ANOVA analyses $p$ values were obtained by post hoc tests used to compare every mean or pair of means (Tukey's & Sidak's) or to compare every mean to a control sample (Dunnet's). Data are represented as means ± standard deviations. ns: not significant, $*p < 0.05$, $**p < 0.01$, $***p < 0.001$, $****<0.0001$.

All representative experimental data (e.g. Western blots, autoradiography, in-gel fluorescence, and electron microscopy analysis) was repeated independently with equivalent results for a minimum of three biologically independent experiments.

## Reporting summary

Further information on research design is available in the Nature Portfolio Reporting Summary linked to this article.

# Data availability

The authors declare that all data supporting the findings of this study are available within the paper, the Supplementary Information and Supplementary Data Source files. Other specific enquiries and data sets are available upon reasonable request.

Further details on materials, methods and computational analysis used in this study can be found in Supplementary Information.

The Mass Spectrometry data sets (processed and RAW data) generated in this study have been deposited in the Mendeley data database and are available at:

Mesquita, Francisco (2022), "Sandoz, P._etal_2022_MassSpecDataSource", Mendeley Data, V1, https://doi.org/10.17632/cx28dr9t22.1 [https://data.mendeley.com/datasets/cx28dr9t22]. Source data are provided with this paper.

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

## Acknowledgements

We thank Dr. Masaki Fukata for the ZDHHC-myc and ZDHHC16-FLAG plasmids; Dr. Ramanujan Hegde for the TRAPα antibody and Dr. Maurizio Molinari for the calnexin antibody. This work was supported in part using the resources and services of the BioEM and PTPSP Research Core Facilities at the School of Life Sciences and the and ISIC-MS facility at the School of Basic Sciences from EPFL, in particular Marie Croisier and Stéphanie Clerc, Thierry Laroche from BioEM; Laurence Durrer and Soraya Quinche from PTPSP; Natalia Galisova from ISIC-MS. The research leading to these results received funding from the European Research Council under the European Union's Seventh Framework Programme (FP/2007-2013)/ERC Grant Agreement no. 340260 - PalmERa. This work was also supported by grants from the Swiss National Centre of Competence in Research (NCCR) Chemical Biology (to G.v.d.G.) and the Swiss SystemsX.ch initiative evaluated by the Swiss National Science Foundation (LipidX) (to G.v.d.G. and to V.H.).

## Author contributions

Conceptualization, P.A.S., R.A.D.E., F.S.M., L.A., V.H., and F.G.v.d.G.; investigation, P.A.S., R.A.D.E., L.A. L.A.A., G.S., C.M., S.H., B.K., K.H., G.K., and F.S.M.; funding acquisition, V.H. and F.G.v.d.G.; writing—original draft, P.A.S., R.A.D.E., F.S.M., L.A., V.H., and F.G.v.d.G.; writing—review and editing, P.A.S., R.A.D.E., F.S.M., L.A., V.H. and F.G.v.d.G.; resources, P.A.S., S.H., L.A., and B.K.

## Competing interests

The authors declare no competing interests.
