## [Peer Review File · Nature Communications]

REVIEWER COMMENTS

Reviewer #1 (Remarks to the Author):

Using a series of established (3H-palmitate, hydroxylamine-dependent PEGylation) and novel (hydroxylamine-dependent fluorescent alkylation) assays of protein S-acylation the paper first demonstrates that the vast majority of CLIMP-63 in cells and tissues is singly acylated in position C100. Using knockout and silencing a functional relationship between zDHHC6, zDHHC2 and zDHHC5 and CLIP-63 palmitoylation is established. These zDHHC-PATs are also found to co-localize with CLIMP-63. A combination of fractionation and labelling approaches identifies multiple populations of CLIMP-63 in the cell: palmitoylated ER resident, non-palmitoylated surface membrane resident, and palmitoylated lipid raft resident. Protein and palmitoylation half-lives are measured and confirm the concept that palmitoylated CLIMP-63 is stabilized and retained in the ER. APT2 is identified as the CLIMP63 depalmitoylating enzyme using genetic and pharmacological approaches.

Using the measured half-lives of CLIMP-63 protein and CLIMP-63 palmitoylation the authors go on to build a mathematical model of CLIMP-63 in the cell. The failure of this model to accurately describe CLIMP-63 behavior leads them to investigate other aspects of CLIMP-63 behavior, in particular oligomerization. The protein is found to self-associate into trimers and higher-order oligomers, most likely dimers of trimers. Refinement of the model correctly predicts CLIMP-63 behavior in the cell, but doesn't appear to replicate experimental findings concerned with the impact of zDHHC siRNA on CLIMP-63 acylation (see Major point 4).

In the final third of the manuscript, the authors evaluate the importance of zDHHC6 mediated CLIMP-63 acylation for ER assembly and morphology. Using elegant loss (cys mutant) and gain (additional S-acylation site) of function mutations of CLIMP-63 the authors demonstrate that reducing CLIMP-63 acylation reduces ER density, while increasing CLIMP-63 acylation expands the ER. A novel imaging approach, persistent homology analysis, is developed to quantify the changes in ER anatomy induced by zDHHC6 acylation of CLIMP-63.

Overall this is a well-conducted study that defines the pathways controlling CLIMP-63 acylation and diacylation, and unequivocally establishes a role for CLIMP-63 acylation in the assembly and anatomy of the endoplasmic reticulum. As such it represents an important advance in understanding cell biology. Most conclusions are well supported by the data presented – with the exception of my reservation about the performance of the model of CLIMP-63 behavior.

Major

1. The PEGylation assays presented in Figure 1b clearly show a bandshift for CLIMP-63 (endogenous) and HA-CLIMP-63 (transfected) after treatment with hydroxylamine and PEG maleimide. However the immunoreactive band has 'disappeared' from the sample that is treated with hydroxylamine but not with PEG (central lane). Why? It is present in C100A HA-CLIMP63, calnexin and TRAPa blots.
2. The data presented in Fig 2 suggest a 'small proportion' (Fig 2f) of CLIMP-63 localizes to the cell surface, with the 'majority' trapped in the ER. A role for zDHHC6 is unequivocally established in the siRNA experiments. Silencing zDHHC6 reduces CLIMP-63 palmitoylation by 80% (zDHHC2 by 30%, zDHHC5 by 40%). Do these numbers really 'add up'? If only a small fraction of CLIMP-63 leaves the ER, how can zDHHC5, living at the end of the secretory pathway, be responsible for palmitoylating 40% of the CLIMP-63 in the cell? When zDHHC6 is silenced, only 20% of CLIMP-63 is palmitoylated – so does this mean that the amount palmitoylated by zDHHC5 is reduced? Since more CLIMP-63 leaves the ER when zDHHC6 is silenced, wouldn't you expect the contributions of zDHHC2 and zDHHC5 to be increased?
3. Fig 3 – are the quoted half-lives in panel j significantly different from each other?
4. Fig 4l – the refined model based on CLIMP-63 oligomerization is demonstrated to correctly predict that CLIMP-63 remains largely confined to the ER, but the oligomer model presented in this panel appears to suggest almost no impact of zDHHC6 siRNA on CLIMP-63 S-acylation (minor increase in E0, no change in H2), which contradicts the experimental data in Fig 2b & 2c (80% decrease in CLIMP-63 acylation with zDHHC6 silencing).

Minor

1. Figure 1 and Supplementary Figure 1 both include the phrase 'input corresponds to 5% of final volume' in the legend. What does this mean?
2. Figure 1d: the '-' and '+' denoting the presence of PEG maleimide above the lanes are the wrong way round.
3. Supplementary 1b – scale bar is not visible but referred to in the legend
4. Supplementary 1d: can the authors re-examine this? I don't think the schematic shows the process correctly. The box called 'immunoprecipitation' on the right suggests the sample is split AFTER being treated with hydroxylamine. Surely only the left hand 'arm' of the assay in this box is treated with HA? It would help if the labels 'A' and 'B' that are used below the WB and Fluo images in the left box could be also added to the two samples in the 'immunoprecipitation' box. In addition, the 'WB' cartoon in the box on the left should surely show identical amounts of the protein of interest being immunoprecipitated? As is indeed the case in Fig 1c?
5. Fig 2h – the DRM fractionation is rather hard to follow. This could be improved with a schematic highlighting what fractions 1-6 actually represent experimentally.
6. Line 265 'and equal distribution' – correct to 'an'?
7. Supplementary Fig 2 and Figure 4 – the legends are cut off by the text box boundaries
8. Figure 7 – why switch to using C100S CLIMP-63 instead of C100A?
9. Line 573 – tuned not tunned

Reviewer #2 (Remarks to the Author):

This is a very interesting paper that characterizes the acylation of CLIMP-63 an ER shaping protein associated with the formation of ER sheets. Overall, a tour de force that encompasses proteomic analysis of acylation, mathematical modeling and correlative light electron microscopy that brings new insight into our understanding of the CLIMP-63 life cycle and its role in shaping ER morphology. The idea that stable CLIMP-63 complexes form in the ER and regulate ER morphology is very interesting however I do have some concerns with respect to interpretation of the data presented.

1. The identification of ZDHHC6 as the ER-localized palmitoyl transferase is interesting and well supported. Concerns are associated with the use of shRNA knockdown and arguments that incomplete knockdown may result in basal and functional levels of palmitoylation of CLIMP-63. Together with discrepancies with behavior of the C100 CLIMP-63 mutant, this argues that to obtain definitive understanding of ZDHHC6 palmitoylation of CLIMP-63, a CRISPR/Cas knockout cell line would be required.
2. I found the proposed role of CLIMP-63 palmitoylation on CLIMP-63 surface expression confusing. On the one hand ZDHHC6 siRNA knockdown increases surface expression, presumably by reducing palmitoylation and preventing ER retention. On the other, mutating the C100 palmitoylation site prevents palmitoylation and surface expression. How do the authors reconcile these discordant results? The idea that the depalmitoylated form is taken to the surface and then palmitoylated by ZDHHC2 and 5 isn't very convincing. Further "our observation that only non-acylated CLIMP-63 exits the ER "(line 311-312) appears to be based on the siZDHHC6 knockdown experiment, yet the authors argue that there is 10% residual ZDHHC6 activity after knockdown which is sufficient to palmitoylate the H2 complex (line 332-333). If so why not E1? As such the model in 4I is compelling with the exception of whether E0 exclusively goes to the cell surface, as the C100 mutant should be exclusively E0. The argument for such a mechanism requires clearer support and perhaps more definitive results with a ZDHHC6 knockout line.
3. A critical control to exclude an indirect effect of ZDHHC6 on CLIMP-63 distribution would be to test the effect of siZDHHC6 (or better ZDHHC6 KO) on C100 surface expression. Similarly an important control would be whether ZDHHC6 overexpression or siRNA/KO affects lifetime of the C100 mutant. Also, does ML349 treatment (Fig 3f) increase surface CLIMP-63 in the presence of siZDHHC6 or siZDHHC2?
4. For the stability assays in figure 3 H-J, the half life analysis does not really represent the plots. There is clearly a more labile population and a more stable population with palmitoylation is more specifically affecting the labile population and most dramatically for the C100 mutant. Is this labile

population the trimers and does palmitoylation stabilize them by enabling their surface delivery? Is the long lived form the higher order complexes? How is the labile population degraded – ERAD/proteasome?

5. The mathematical modeling is very interesting and informative. It would be important to more clearly identify *in silico* from real experiments in the figures and in the text.

6. The effect of expression of the C100 mutant on ER network organization (Supp fig 6 gh) isn't obvious and there are still a lot of sheets. Similarly the reported ER expansion isn't always obvious either. Is the ZDHHC6 effect on ER morphology in Fig 6A C100A dependent? And is the very obvious CLIMP-63-CC effect on the ER ZDHHC6-dependent? The CLEM is very nice.

Supp fig 4A – where is the monomer in the graphs?

Reviewer #3 (Remarks to the Author):

This study, by Sandoz et al, investigates the regulation and cellular distribution of the ER protein Climp-63 by a reversible post-translational modification: S-acylation (S-palmitoylation). The group has a long-standing interest and expertise on Climp-63 and S-palmitoylation. In the current study, they validate and advance their previous findings that human Climp-63 protein can be reversibly palmitoylated at Cysteine100.

The investigators additionally show that the ER localized acyltransferase ZDHHC6 palmitoylates Climp-63 which plays a role on its retention in the ER. This modification can be reversed by Protein Acyl Thioesterase 2 (APT2). They further investigate the half-lives of different forms and states of Climp-63 by taking advantage of mathematical modeling and show that higher-order assemblies of Climp-63 protects it from depalmitoylation, thus increasing its stability in the ER. Finally, they investigate the impact of Climp-63 palmitoylation on ER architecture and abundance.

Overall, the experiments, especially radioactive labeling, and pulse-chase experiments, are well conducted with appropriate controls. The manuscript is well written, and clearly deserves publication with some points to consider by the authors (described below).

Major points:

1) The Acyl-Rac or other similar assays evaluate the level of modified cysteines overall. Authors state in line 150 that "in various mouse organs, the majority of Climp-63 molecules are lipid-modified by S-acylation". However, this methodology does not exclude other types of reversible native cysteine modifications (e.g. S-sulfinylation, etc.). This should be rephrased in the text. Also, in the current study, all of the evidence regarding Climp-63 S-acylation is shown in human Climp-63 protein. Mouse Climp-63 is slightly different than human Climp-63. Do authors have evidence (mutant) that Cys79 of mouse Climp-63 is similarly modified compared to Cys100 of human Climp-63? These points can be clarified in the manuscript.

2) The authors show no difference in Climp-63 expression when ZDHHC6 is overexpressed in Supp. Fig. 2c and only a 20% increase in Fig. 5j. However, Fig. 2d shows a clear increase in Climp-63 expression and distribution (top right image vs images below). Can authors clarify this variation please?

3) In Fig.2e, is the quantification done in a 3D Z-stack or in a single confocal plane? This type of analysis might suffer bias if it's done on a single optical plane. It might be better to normalize it by cell area and clearly describe in the manuscript.

4) Where is endogenous ZDHHC6 localized in the ER? Does it have a preferential localization to rough ER sheets, where it can modify both Climp-63 and Calnexin? Perhaps, co-staining with endogenous Sec61-beta and/or Rtn4A would be insightful.

5) In Supp. Fig. 1b, is it possible to add C100A mutant staining as well? Does this mutation change

its distribution in the ER?

6) In Supp. Fig. 6h, it is not possible qualitatively to reach authors' conclusion on the effects of the mutants on ER shape. Could the authors provide better images or add quantification?

7) In Fig. 6a and Supp. Fig. 6c, the ZDHHC6-myc expression looks like it's aggregated, and this is different than the staining pattern of ZDHHC6-myc in Fig. 7a. Could you please clarify this difference?

8) It's difficult to evaluate ER expansion or ER morphology from the images presented in Supp Fig. 6c. In Fig 6a, it seems like cells overexpressing ZDHHC6 contains vacuole type of regions? Is this how the investigators judged these structures as well? If so, what may these structures be? If the authors mean "ER dilation" by the term ER expansion, perhaps staining of a luminal ER protein would be helpful to show the ER border.

9) For Supp. Fig. 6f, I recommend checking XBP1 splicing (sXBP1 / tXBP1 ratio) and EIF2alpha phosphorylation to evaluate UPR, since these factors are more relevant for ER expansion and dilation.

10) In Fig. 7a, why did the authors switch to C100S mutant, and not use the C100A? It may be useful for the readers to clarify.

Minor:

1) In Fig.1c, the top blot doesn't represent the quantification shown below it. It would be better to include a clearer blot.

2) In line 235, citation reference "2" doesn't explain the statement.

3) Citation 50 is published, please update the reference (PMID: 35264794).

Reviewer #4 (Remarks to the Author):

This study attempts to reveal the lifecycle of CLIMP-63: a key protein for ER morphology, applying a combination of experimental methods and mathematical modeling. I (theoretical researcher) avoid judging the biological significance of the finding but comment on the mathematical modeling. This study seems to use the mathematical models appropriately and to address the outcome from them adequately. However, lacking some information about models and several confusing statements make the description of the model results unclear.

First, it should be clarified what the authors call "model". On lines 315-316, the authors state "a population of models" and the following sentence contains "100 sets of parameters". Also, the caption of Fig. 4k notes "100 model simulations". At the same time, the latter paragraphs of the manuscript contain the term "the model" (lines 322, 328, 364, and so on). It seems that all indicate the same thing. If true, it is confusing. Or, if "the model" indicates a different thing, I cannot figure it out.

The description of the models is scattered across the manuscript. It is not easy to grasp the structure of the models. The first (monomer) model is only shortly described in the manuscript (lines 261-264). Variables of the second (E+H) model are described in the manuscript (line 310), the reaction and the transportation network are displayed in Fig. 4i without the information of enzymes, the enzymatic reaction scheme is noted in the supplementary text, and the list of parameters is in Supplementary Fig. 5c. Explicit indications of model equations, which contain all information, will help readers understand the model structures. The variable E^0_{CP} shown only in Fig. 4i should be mentioned in the manuscript.

The procedures of the parameter estimation and the global sensitivity analysis should be described

including the determination of the fitness function.

The explanation and the usage of persistent homology are appropriate.

Point-by-point reply

Reviewer #1

We thank the reviewer for the excellent and complete summary of our work. We are pleased that the work for the most part convincingly shows a role for CLIMP-63 acylation in the assembly and anatomy of the ER and that it represents an important advance in understanding cell biology.

Major specific comments

1. *The PEGylation assays presented in Figure 1b clearly show a bandshift for CLIMP-63 (endogenous) and HA-CLIMP-63 (transfected) after treatment with hydroxylamine and PEG maleimide. However, the immunoreactive band has 'disappeared' from the sample that is treated with hydroxylamine but not with PEG (central lane). Why? It is present in C100A HA-CLIMP63, calnexin and TRAPa blots.*

We have now replaced the blots with those from experiments where CLIMP-63 is visible. For technical reasons we do not fully understand, in some experiments CLIMP-63 is not seen in the + HA -PEG samples.

2. *The data presented in Fig 2 suggest a 'small proportion' (Fig 2f) of CLIMP-63 localizes to the cell surface, with the 'majority' trapped in the ER. A role for zDHHC6 is unequivocally established in the siRNA experiments. Silencing zDHHC6 reduces CLIMP-63 palmitoylation by 80% (zDHHC2 by 30%, zDHHC5 by 40%). Do these numbers really 'add up'? If only a small fraction of CLIMP-63 leaves the ER, how can zDHHC5, living at the end of the secretory pathway, be responsible for palmitoylating 40% of the CLIMP-63 in the cell? When zDHHC6 is silenced, only 20% of CLIMP-63 is palmitoylated – so does this mean that the amount palmitoylated by zDHHC5 is reduced? Since more CLIMP-63 leaves the ER when zDHHC6 is silenced, wouldn't you expect the contributions of zDHHC2 and zDHHC5 to be increased?*

These are all very valid questions. The apparent discrepancy comes from the fact that in Fig. 2f, surface biotinylation, or all western-based experiments, monitor the total CLIMP-63 population. In contrast ³H-palmitate labelling only reveals the population that undergoes S-acylation during the pulse time, 2 hrs in Fig. 2c. Since the vast majority of cellular CLIMP-63 is palmitoylated in control cells, the proportion of CLIMP-63 available for palmitoylation is small, and that is the one revealed in Fig. 2c. We now mention that indeed the results of ³H-palmitate labelling “don't add up” as the reviewer mentions, and that this is probably because the amount of CLIMP63 available for palmitoylation changes when we silence one or the other ZDHHC enzymes.

As suggested by the reviewer, when zDHHC6 is silenced, the contributions of zDHHC2 or zDHHC5 is expected to be increased, since the pool of CLIMP63 with a “free” cysteine might be higher. Consistently we find an increased presence of CLIMP-63 in the detergent resistant membrane fraction, Fig. 2h, which corresponds to surface CLIMP-63 palmitoylated by zDHHC 2 or 5.

3. Fig 3 – are the quoted half-lives in panel j significantly different from each other?

We have now determined the half-life of CLIMP-63 for each individual experiment (Fig 3h and 3j) in order to provide standard deviations of the half-lives and statistical significances of the differences. All half-lives were statistically different from each other, with the exception of zDHHC2 silencing which did not alter the global half-life of endogenous CLIMP-63, as expected since the bulk of CLIMP-63, which is in the ER, never reached the plasma membrane and is thus not exposed to ZDHHC2/5 modification.

4. *Fig 4l – the refined model based on CLIMP-63 oligomerization is demonstrated to correctly predict that CLIMP-63 remains largely confined to the ER, but the oligomer model presented in this panel appears to suggest almost no impact of zDHHC6 siRNA on CLIMP-63 S-acylation*

(minor increase in E^0 , no change in H^2), which contradicts the experimental data in Fig 2b & 2c (80% decrease in CLIMP-63 acylation with zDHHC6 silencing).

We have extensively rewritten the description of the model to make it clearer and more accessible. We have also clarified the point mentioned above: ^{35}S Cys/Met experiments document on the acylation that occurs during the 2 h labelling period and which can only happen on the “non-acylated” CLIMP-63 population. It does not provide information on the percentage of total CLIMP-63 that is acylated in the siZDHHC6 silenced cells. In the model, the residual ZDHHC6 activity in siZDHHC6 silenced cells was set to 10%. Therefore, acylation can still occur. The 90% reduced acylation activity leads to an increase in E^0 , which can exit the ER, leading to an increased plasma membrane population which appears as E^1_{PM} , since ZDHHC2&5 can operate. The fact that ZDHHC6-mediated palmitoylation can still occur due to the residual activity, leads to the accumulation over time of the most stable species which is H^2_{ER} . We also analysed the distribution of the C100A mutant, which in agreement with the experimental observations, accumulates in the H^0 form in the ER. It is barely present at the plasma membrane, not because it does not arrive there, but because it does not stay there. Acylation at the surface is indeed required to increase the dwell time of CLIMP-63 at the plasma membrane. This is now all more explicit in the text. Also, we have moved the description of the species distribution of C100A, next to that of WT, which makes things much clearer. So, thank you for raising this point.

Minor

1. *Figure 1 and Supplementary Figure 1 both include the phrase ‘input corresponds to 5% of final volume’ in the legend. What does this mean?*

We thank the reviewer for this comment which allowed us to correct this point, which was incorrect. The total amount of protein in the lanes of input and the two condition in Figure 1 is the same. In Suppl. Fig.1, the amount of protein in the input is the same as the amount of protein on which the pull-down was performed. This has now been clarified in the legends.

2. *Figure 1d: the ‘-’ and ‘+’ denoting the presence of PEG maleimide above the lanes are the wrong way round.*

The signs have been corrected.

3. *Supplementary 1b – scale bar is not visible but referred to in the legend*

The scale bar has been added.

4. *Supplementary 1d: can the authors re-examine this? I don’t think the schematic shows the process correctly. The box called ‘immunoprecipitation’ on the right suggests the sample is split AFTER being treated with hydroxylamine. Surely only the left hand ‘arm’ of the assay in this box is treated with HA? It would help if the labels ‘A’ and ‘B’ that are used below the WB and Fluo images in the left box could be also added to the two samples in the ‘immunoprecipitation’ box. In addition, the ‘WB’ cartoon in the box on the left should surely show identical amounts of the protein of interest being immunoprecipitated? As is indeed the case in Fig 1c?*

We thank the reviewer for picking this up. There was indeed an error in the figure, which was not present in the BioRxiv version of the manuscript and which we have now corrected.

5. *Fig 2h – the DRM fractionation is rather hard to follow. This could be improved with a schematic highlighting what fractions 1-6 actually represent experimentally.*

A scheme was added in Suppl. Fig. 2g.

The three following corrections have been made:

6. Line 265 'and equal distribution' – correct to 'an'?
 7. Supplementary Fig 2 and Figure 4 – the legends are cut off by the text box boundaries
 9. Line 573 – tuned not tunned
8. Figure 7 – why switch to using C100S CLIMP-63 instead of C100A?

This is historical. At some point we thought that serine mutations would be better than Alanine mutations, but in practice we have never seen a difference on any protein, nor has anyone in the field reported that the residue to which cysteines are mutated (alanine or serine) make a difference.

Reviewer #2

We are delighted that the reviewer appreciates the work and the findings described in our manuscript!

1. The identification of ZDHHC6 as the ER-localized palmitoyl transferase is interesting and well supported. Concerns are associated with the use of shRNA knockdown and arguments that incomplete knockdown may result in basal and functional levels of palmitoylation of CLIMP-63. Together with discrepancies with behavior of the C100 CLIMP-63 mutant, this argues that to obtain definitive understanding of ZDHHC6 palmitoylation of CLIMP-63, a CRISPR/Cas knockout cell line would be required.

We believe there might have been some confusion in the paper (that we have now clarified in the text), and therefore to the reviewer (as also apparent from point 2 below):

- While ZDHHC6 was silenced with siRNA in many of the experiments in the paper, we did generate a ZDHHC6 KO cell line, which was used for some key experiments in the paper and we have now added some more, described in the next points (Figs 2a,d,e, Supp. 3f,g 4o and 6f,g).
- It is for CLIMP-63 that we used an shRNA cell line in which the endogenous CLIMP-63 was undetectable by western blotting. Given how abundant CLIMP-63 is, the shRNA cells were considered adequate for our purposes. However, we have now added some experiments in which we have silenced CLIMP-63 in the shRNA CLIMP-63 cells, just to ensure there was as little residual CLIMP-63 as possible.

2. I found the proposed role of CLIMP-63 palmitoylation on CLIMP-63 surface expression confusing. On the one hand ZDHHC6 siRNA knockdown increases surface expression, presumably by reducing palmitoylation and preventing ER retention. On the other, mutating the C100 palmitoylation site prevents palmitoylation and surface expression. How do the authors reconcile these discordant results? The idea that the depalmitoylated form is taken to the surface and then palmitoylated by ZDHHC2 and 5 isn't very convincing. Further "our observation that only non-acylated CLIMP-63 exits the ER" (line 311-312) appears to be based on the siZDHHC6 knockdown experiment, yet the authors argue that there is 10% residual ZDHHC6 activity after knockdown which is sufficient to palmitoylate the H2 complex (line 332-333). If so why not E1? As such the model in 4I is compelling with the exception of whether E0 exclusively goes to the cell surface, as the C100 mutant should be exclusively E0. The argument for such a mechanism requires clearer support and perhaps more definitive results with a ZDHHC6 knockout line.

We have extensively rewritten the text to clarify the information that is provided by experiments on the C100A mutant. We also provide a more detailed description of the model to make it more accessible. Finally, we have made more explicit what the model "tells" us.

We have also performed additional experiments on the ZDHHC6 KO cell line. More specifically:

- As requested by Reviewer 3, we now show IF of C100A in the very beginning (Fig 1b) of the paper, which shows that it mostly localises to the ER, as WT CLIMP-63. Blue Native gel analysis moreover shows that C100A has a similar quaternary assembly into E (trimer) and H (higher order), as WT.
- The palmitoylation of CLIMP-63 at the cell surface/endosomes has been extensively studied (Planey et al., Mol.Cell Biol. 2009, Sada et al., Sci. Signal 2019). We

therefore did not further address this point. Our findings are however consistent with these published observations.

- We have reworded lines 332-333. The reviewer is absolutely correct: residual activity will lead to E¹ and subsequently H².
- The data and the model, in full agreement with one another, lead to the following conclusion: two mechanisms are each sufficient to retain CLIMP-63 in the ER: S-acylation and higher order assembly into H. Thus, only E⁰ can exit the ER and reach the plasma membrane. There, acylation leads to a 4-fold increase in half-life, leading to the accumulation of E¹_{PM}, with E⁰_{PM} being degraded. The C100A therefore does not prevent surface expression *per se*, but C100A is barely present at the surface because it is internalized.

Since in the ER, E associates into H, very little exits the ER, even for C100A, or WT in ZDHHC6 KO cells.

3. *A critical control to exclude an indirect effect of ZDHHC6 on CLIMP-63 distribution would be to test the effect of siZDHHC6 (or better ZDHHC6 KO) on C100 surface expression.*

We thank the reviewer for suggesting this control. We monitored the surface expression of C100A both in siZDHHC6 cells, and additionally tested the silencing of ZDHHC 2 and 5. The surface expression of C100A remained essentially identical under all conditions.

The text therefore now includes: *“To rule out general effect of ZDHHC6 silencing on biosynthetic trafficking, we quantified the presence of CLIMP-63 C100A mutant, which also localizes mostly to the ER (Sup. Fig. 1b), under different conditions. While the amount of C100A at the plasma membrane was far lower than that of WT CLIMP-63 in control cells (Fig. 2i), its surface abundance was insensitive to ZDHHC2 or 6 silencing (Fig. 2i and Supp Fig 2f).”*

Similarly an important control would be whether ZDHHC6 overexpression or siRNA/KO affects lifetime of the C100 mutant.

We have performed ³⁵S Cys/Met labelling to analyse the decay of C100A under the following conditions:

* ZDHHC6 KO with or without silencing of endogenous CLIMP-63 and shCLIMP-63 cells all expressing C100A mutant.

The decay curves under all conditions were very similar (Supp Fig 3g), indicating that C100A turnover rate is insensitive to the presence of ZDHHC6.

Also, does ML349 treatment (Fig 3f) increase surface CLIMP-63 in the presence of siZDHHC6 or siZDHHC2?

We monitored the effects of ML349 on the levels of surface CLIMP-63, WT and C100A, by surface biotinylation, under various conditions, in both control and ZDHHC6 KO cells. These are now described in the text as follows:

“We next investigated the effect of ML349 on the amount of CLIMP-63 at the cell surface. ML349 led to an almost 4-fold increase of endogenous CLIMP63 at the plasma membrane (Fig. 3f), while as expected ML349 did not affect the amount of surface C100A mutant (Supplementary Fig. 3e). We have previously shown that inhibiting APT2 leads to rapid degradation of ZDHHC6⁴⁰. The observed ML349-induced increase of CLIMP-63 at the cell surface could thus be partly due to an indirect effect on ZDHHC6. We therefore repeated the experiment in ZDHHC6 KO cells. ML349 still had an effect on endogenous surface CLIMP63 (Supplementary Fig. 3f), although lower, arguing for an effect of ML349 on ZDHHC6. The effect of ML349 on the surface expression of CLIMP-63 was essentially lost when ZDHHC2, 5 or both were silenced in Ctrl cells (Fig. 3f) and in ZDHHC6 KO cells (Supplementary. Fig. 3f). These observations indicate that surface CLIMP-63, once S-acylated by ZDHHC 2 or 5, can undergo de-acylation by APT2, and that this de-acylation at the plasma membrane leads to a decrease of surface CLIMP-63, presumably due to retrieval of non-acylated CLIMP-63 by endocytosis³³.”

4. *For the stability assays in figure 3 H-J, the half life analysis does not really represent the plots. There is clearly a more labile population and a more stable population with palmitoylation is more specifically affecting the labile population and most dramatically for the C100 mutant. Is this labile population the trimers and does palmitoylation stabilize them by enabling their surface delivery? Is the long lived form the higher order complexes? How is the labile population degraded – ERAD/proteasome?*

We fully agree with the comments of the reviewer. The half-lives that are provided based ³⁵SCys/Met decay curves are “apparent half-lives”, i.e. when 50% of the protein is degraded. Such curves only provide the real half-life when studying a single species, but as the reviewer underlines, there is at least one labile and one stable population. The complexity of this curve can only be dissected once the model allows to deconvolute this decay curve, providing the estimated half-lives of all the species (or populations) as shown in figure 5f. We now mention when describing Fig. 3 that the decay curves appear biphasic and that this will be addressed later in the paper.

So, all the questions of the reviewer are highly valid, but can only be answered when arriving at figure 5.

The question of the degradation route of the labile population is very interesting, because it is not via the proteasome. If it had been by the proteasome, we could have just mentioned it. But it is not, and we don't yet know how E⁰_{ER} gets degraded.

5. *The mathematical modelling is very interesting and informative. It would be important to more clearly identify in silico from real experiments in the figures and in the text.*

We now provide more details in the text on the model, we clarified what are predictions and what are experimental results, both in the text and in the figures.

6. *The effect of expression of the C100 mutant on ER network organization (Supp fig 6 gh) isn't obvious and there are still a lot of sheets. Similarly the reported ER expansion isn't always obvious either. Is the ZDHHC6 effect on ER morphology in Fig 6A C100A dependent? And is the very obvious CLIMP-63-CC effect on the ER ZDHHC6-dependent? The CLEM is very nice.*

The structure of the ER is indeed complex. Irrespective of sheets or tubules, the perinuclear area is always extremely dense. Moreover, the phenotypes are never, all or none. So there are indeed still sheets in the CLIMP-CC-expressing cells. Fig. 6a shows the effect of the overexpression of WT CLIMP-63. Fig. 6c shows the quantification comparing over expression of WT and C100A. Therefore, indeed the effect on ER morphology shown in Fig. 6a is C100A dependent as we now clear underline.

We have now added an experiment that shows that the effect of the CC mutant indeed requires the activity of ZDHHC6 (Fig. 6j, k).

7. *Supp fig 4A – where is the monomer in the graphs?*

We have improved the figure lines and colours to make each species more visible.

Reviewer #3

We are pleased that the reviewer found the study interesting, the manuscript clearly written and worthy of publication. We have addressed all the major points:

1) *The Acyl-Rac or other similar assays evaluate the level of modified cysteines overall. Authors state in line 150 that “in various mouse organs, the majority of Climp-63 molecules are lipid-*

modified by S-acylation". However, this methodology does not exclude other types of reversible native cysteine modifications (e.g. S-sulfinylation, etc.). This should be rephrased in the text.

We have rephrased the text to refer to *thioester bond mediated modifications*.

Also, in the current study, all of the evidence regarding Climp-63 S-acylation is shown in human Climp-63 protein. Mouse Climp-63 is slightly different than human Climp-63. Do authors have evidence (mutant) that Cys79 of mouse Climp-63 is similarly modified compared to Cys100 of human Climp-63? These points can be clarified in the manuscript.

As requested by the reviewer we have analysed mouse CLIMP-63. We now show that mouse CLIMP-63 is S-acylated and that it occurs on Cys-79. This is mentioned in the text and shown in Supplementary Fig. 1g, h.

2) The authors show no difference in Climp-63 expression when ZDHHC6 is overexpressed in Supp. Fig. 2c and only a 20% increase in Fig. 5j. However, Fig. 2d shows a clear increase in Climp-63 expression and distribution (top right image vs images below). Can authors clarify this variation please?

When looking at cell populations using western blots, as for figs. Suppl. 2c and 5j, the increase was around 20%. There is however significant variability amongst cells which is visible by fluorescence microscopy, but then averaged out when quantifying many cells. We have however changed the figure, choosing more average looking cells.

3) In Fig.2e, is the quantification done in a 3D Z-stack or in a single confocal plane? This type of analysis might suffer bias if it's done on a single optical plane. It might be better to normalize it by cell area and clearly describe in the manuscript.

We agree with the reviewer and have now normalised our previous data to the cell area measured by CLIMP-63 distribution. Normalisation did not change the previous conclusions. We have also detailed the quantification of the PLA assay in the methods section.

4) Where is endogenous ZDHHC6 localized in the ER? Does it have a preferential localization to rough ER sheets, where it can modify both Climp-63 and Calnexin? Perhaps, co-staining with endogenous Sec61-beta and/or Rtn4A would be insightful.

These are very relevant questions. Unfortunately, as with most ZDHHC enzymes, no antibodies that work by IF are available. Endogenous ZDHHC6 can be seen by western blot, when sufficiently abundant. ZDHHC enzymes are low copy proteins, probably less than 1000 copies per cell, according to some quantitative proteomics studies. Overexpressed ZDHHC6 essentially stains the entire ER (Fig 2d and 6j) but is unlikely to reveal localization of the endogenous protein.

So at this stage, we cannot answer the interesting question regarding potential specific ER domain localization of ZDHHC6.

We have however published FRAP experiments on ectopically expressed ZDHHC6 and found that it is highly mobile in the ER, the aim was then to show how it could encounter calnexin (Dallavilla et al, PLoS Comp. boil. 2016).

5) In Supp. Fig. 1b, is it possible to add C100A mutant staining as well? Does this mutation change its distribution in the ER?

We have now added imaging of endogenous CLIMP-63 in Ctrl cells, WT and C100A HA-CLIMP-63 in shCLIMP-63 cells, showing that the distribution is similar in all cases. Adding the C100A IF staining very early on in the manuscript has been very beneficial, clarifying from the start that acylation is not an absolute requirement for ER localization of CLIMP-63. Thank you.

6) In Supp. Fig. 6h, it is not possible qualitatively to reach authors' conclusion on the effects of the mutants on ER shape. Could the authors provide better images or add quantification?

We now show images that provide a better qualitative view of the effects of the mutants on ER morphology. We now also cite a recent publication that indicates that the CLIMP-63 C100A mutant alters the organisation of ER-mitochondria contact sites (Harada et al. 2020)

7) In Fig. 6a and Supp. Fig. 6c, the ZDHHC6-myc expression looks like it's aggregated, and this is different than the staining pattern of ZDHHC6-myc in Fig. 7a. Could you please clarify this difference?

The fluorescence images in Fig. 7a correspond to RFP-CLIMP-63. Regarding zDHHC6, overexpressed zDHHC6 may lead to the formation of OSERs in some regions of the ER, as illustrated in Suppl. Fig. 7. These are not aggregates but highly ordered ER structures (OSERs), one of which is seen in the FIB-SEM analysis. This is now underlined in the text.

8) It's difficult to evaluate ER expansion or ER morphology from the images presented in Supp Fig. 6c. In Fig 6a, it seems like cells overexpressing ZDHHC6 contains vacuole type of regions? Is this how the investigators judged these structures as well? If so, what may these structures be? If the authors mean "ER dilation" by the term ER expansion, perhaps staining of a luminal ER protein would be helpful to show the ER border.

In cells heavily overexpressing ZDHHC6, the ER indeed has a vacuole-like appearance in normal confocal microscopy due to the concentration of ER in dense structures emanating from the perinuclear region. We agree with the reviewer that expansion is not the ideal term. We now use the "dilated" ER as suggested. We were also intrigued by what these structures might be and this prompted use to perform FIB-SEM. This high-resolution analysis showed that the ER lost its fenestration. So "de-fenestrated" would be the adequate term, as used in the conclusion, but the flow of the paper does not allow us to use the term from the start.

9) For Supp. Fig. 6f, I recommend checking XBP1 splicing (sXBP1 / tXBP1 ratio) and EIF2alpha phosphorylation to evaluate UPR, since these factors are more relevant for ER expansion and dilation.

We have now monitored XBP1 splicing and EIF2alpha phosphorylation upon ZDHHC6 over expression. The data shows that there is a very mild UPR, as compared to tunicamycin treatment. As an additional control, we now show that overexpression of ZDHHC6 does not lead to an inhibition of protein synthesis, we actually even see a small increase. We thank the reviewer for this suggestion.

10) In Fig. 7a, why did the authors switch to C100S mutant, and not use the C100A? It may be useful for the readers to clarify.

This is historical. At some point we thought that serine mutations would be better than Alanine mutations, but in practice we have never seen a difference on any protein, nor has anyone in the field reported that the residue to which cysteines are mutated (alanine or serine) make a difference.

Minor:

1) In Fig. 1c, the top blot doesn't represent the quantification shown below it. It would be better to include a clearer blot.

We have now inverted the colour to make the low intensity band more visible.

The following two citations have been corrected:

2) In line 235, citation reference "2" doesn't explain the statement.

3) Citation 50 is published, please update the reference (PMID: 35264794).

Reviewer #4

We are pleased that the reviewer finds that we have used *the mathematical models appropriately and address the outcome from them adequately*.

1. *First, it should be clarified what the authors call "model". On lines 315-316, the authors state "a population of models" and the following sentence contains "100 sets of parameters". Also, the caption of Fig. 4k notes "100 model simulations". At the same time, the latter paragraphs of the manuscript contain the term "the model" (lines 322, 328, 364, and so on). It seems that all indicate the same thing. If true, it is confusing. Or, if "the model" indicates a different thing, I cannot figure it out.*

We have extensively rewritten and expanded the description of the model to increase clarity and understanding. Also, the supplementary information has been extended.

2. *The description of the models is scattered across the manuscript. It is not easy to grasp the structure of the models. The first (monomer) model is only shortly described in the manuscript (lines 261-264). Variables of the second (E+H) model are described in the manuscript (line 310), the reaction and the transportation network are displayed in Fig. 4i without the information of enzymes, the enzymatic reaction scheme is noted in the supplementary text, and the list of parameters is in Supplementary Fig. 5c. Explicit indications of model equations, which contain all information, will help readers understand the model structures. The variable E^0_{CP} shown only in Fig. 4i should be mentioned in the manuscript.*

These are relevant points. We have included a unified description of the mathematical model to facilitate the understanding and referenced the supplementary information where appropriate.

Enzyme information has been included in the text and legend of Fig. 4i.

We now provide a full rule-based model in the supplementary material. We felt that having it in the text might be confusing to the non-computational reader.

We now mention E^0_{CP} in the text, and specify that it is only a transport step, with no enzymatic component associated.

3. *The procedures of the parameter estimation and the global sensitivity analysis should be described including the determination of the fitness function.*

These are now described in the supplementary information.

REVIEWERS' COMMENTS

Reviewer #1 (Remarks to the Author):

I thank the authors for their detailed and comprehensive responses. The revisions have improved the clarity of the manuscript and strengthened the conclusions.

Minor

Legend to Supplementary Figure 1f currently reads 'whereas B corresponds to the non-cleaved cysteines the non-acylated cysteines'. For clarity, better to only say non-acylated? The schematic in Supplementary 1f is still confusing - in the box on the right it appears to suggest 'cleavage NH₂OH then split the sample' whereas the intended meaning is 'split the sample and cleave half with NH₂OH'. Perhaps you could label the NH₂OH treated arm of the experiment 'A' and the untreated arm 'B'; to match the labelling of the mock western / fluorescent gel.

Line 174 - I suggest you clarify that the lack of 3H-palmitate incorporation into endogenous CLIMP-63 refers to a '2 hour labelling period'. This will avoid confusion that the modelling presented later does not predict reduced steady state palmitoylation of CLIMP-63 when zDHHC6 is silenced.

Figure 4o - missing 'l' in 'levels'

Reviewer #2 (Remarks to the Author):

My previous concerns have been addressed, This represents an interesting and important contribution to the regulation of CLIMP-63 by S-acylation and how that impacts its role in ER organization.

Reviewer #3 (Remarks to the Author):

The authors have responded to the suggestions we have made during the initial review and revised to strengthen their manuscript. There are no further comments.

Reviewer #4 (Remarks to the Author):

All my queries are fulfilled by the revised manuscript.

Point-by-point reply

Reviewer #1 (Remarks to the Author):

We thank the reviewer for his final remarks, all the final issues raised have been corrected or improved.

1 - *Legend to Supplementary Figure 1f currently reads 'whereas B corresponds to the non-cleaved cysteines the non-acylated cysteines'. For clarity, better to only say non-acylated? The schematic in Supplementary 1f is still confusing - in the box on the right it appears to suggest 'cleavage NH₂OH then split the sample' whereas the intended meaning is 'split the sample and cleave half with NH₂OH'. Perhaps you could label the NH₂OH treated arm of the experiment 'A' and the untreated arm 'B; to match the labelling of the mock western / fluorescent gel.*

The Supplementary Figure 1f and correspondent legend have now been corrected and updated taking into account the reviewer's suggestions.

2 - Line 174 - I suggest you clarify that the lack of 3H-palmitate incorporation into endogenous CLIMP-63 refers to a '2 hour labelling period'. This will avoid confusion that the modelling presented later does not predict reduced steady state palmitoylation of CLIMP-63 when zDHHC6 is silenced.

Reference to the length of the pulse period is now mentioned in the text.

3 - *Figure 4o - missing 'l' in 'levels'*

Figure 4o has been corrected

As a final remark, while preparing the data source file with all uncropped image blots we noticed that in Fig 2a the blot of CLIMP-63 IP-fraction was not the corresponding experiment for the autoradiographic image presented. That aspect has been corrected in the image and the data source blots are provided.

We thank all reviewers for the thorough reviews of our work. We are pleased to have addressed each of the raised questions and to provide such an improved version of our work.